# D-Fusion: Direct Preference Optimization for Aligning Diffusion Models with Visually Consistent Samples

**Zijing Hu** [* 1]   **Fengda Zhang** [* 2]   **Kun Kuang** [1]

## Abstract

The practical applications of diffusion models have been limited by the misalignment between generated images and corresponding text prompts. Recent studies have introduced direct preference optimization (DPO) to enhance the alignment of these models. However, the effectiveness of DPO is constrained by the issue of visual inconsistency, where the significant visual disparity between well-aligned and poorly-aligned images prevents diffusion models from identifying which factors contribute positively to alignment during fine-tuning. To address this issue, this paper introduces D-Fusion, a method to construct DPO-trainable visually consistent samples. On one hand, by performing mask-guided self-attention fusion, the resulting images are not only well-aligned, but also visually consistent with given poorly-aligned images. On the other hand, D-Fusion can retain the denoising trajectories of the resulting images, which are essential for DPO training. Extensive experiments demonstrate the effectiveness of D-Fusion in improving prompt-image alignment when applied to different reinforcement learning algorithms.

## 1. Introduction

Diffusion models have made remarkable success in various domains, such as medicine (Xu et al., 2022), robotics (Chi et al., 2024), and 3D synthesis (Poole et al., 2022). Recently, the application of diffusion models in the field of text-to-image generation has gained widespread attention (Ho et al., 2020; Dhariwal & Nichol, 2021). Under the guidance of the given text descriptions, usually called *prompts*, these

---
[*]Equal contribution [1]College of Computer Science and Technology, Zhejiang University, Hangzhou, China [2]College of Computing and Data Science, Nanyang Technological University, Singapore. Correspondence to: Kun Kuang <kunkuang@zju.edu.cn>.

*Proceedings of the $42^{nd}$ International Conference on Machine Learning*, Vancouver, Canada. PMLR 267, 2025. Copyright 2025 by the author(s).

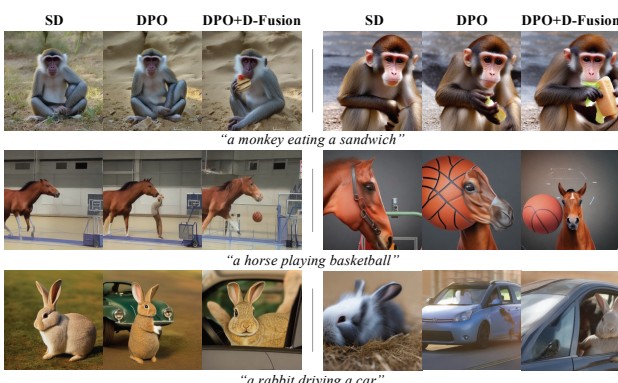

*Figure 1.* (**Misalignment**) Diffusion models (*e.g.*, Stable Diffusion (SD) (Rombach et al., 2022)) often encounter the issue that the generated images do not accurately match the given prompts. Existing RL-based fine-tuning methods (*e.g.*, DPO (Wallace et al., 2023)) have limited effectiveness in improving the alignment. For each set of images above, we use the same seed for sampling.

models transform random noises to corresponding images via a sequential denoising process. However, as shown in Figure 1, diffusion models often encounter the issue of *prompt-image misalignment* (Jiang et al., 2024; Mrini et al., 2024). Prompt-image misalignment refers to the problem that the generated images do not accurately match the given text prompts, which limits the real-world applications of diffusion models.

To address this issue, recent studies have explored incorporating reinforcement learning (RL) algorithms to fine-tune pre-trained diffusion models (Black et al., 2024; Clark et al., 2024; Fan et al., 2023; Wallace et al., 2023; Xu et al., 2023; Yang et al., 2024a;b; Hu et al., 2025). In the paradigm of RL, the step-by-step denoising process of diffusion models is reinterpreted as a *sequential decision-making problem*. In this formulation, the intermediate noisy image at each timestep is regarded as a *state*, while each denoising operation corresponds to an *action*. Among these RL algorithms, direct preference optimization (DPO) stands out for its advantage of eliminating the need for an explicit *reward model*, making it a widely adopted approach (Wallace et al., 2023; Yang et al., 2024a). As illustrated in Figure 2(a), researchers first sample images from the diffusion model with given

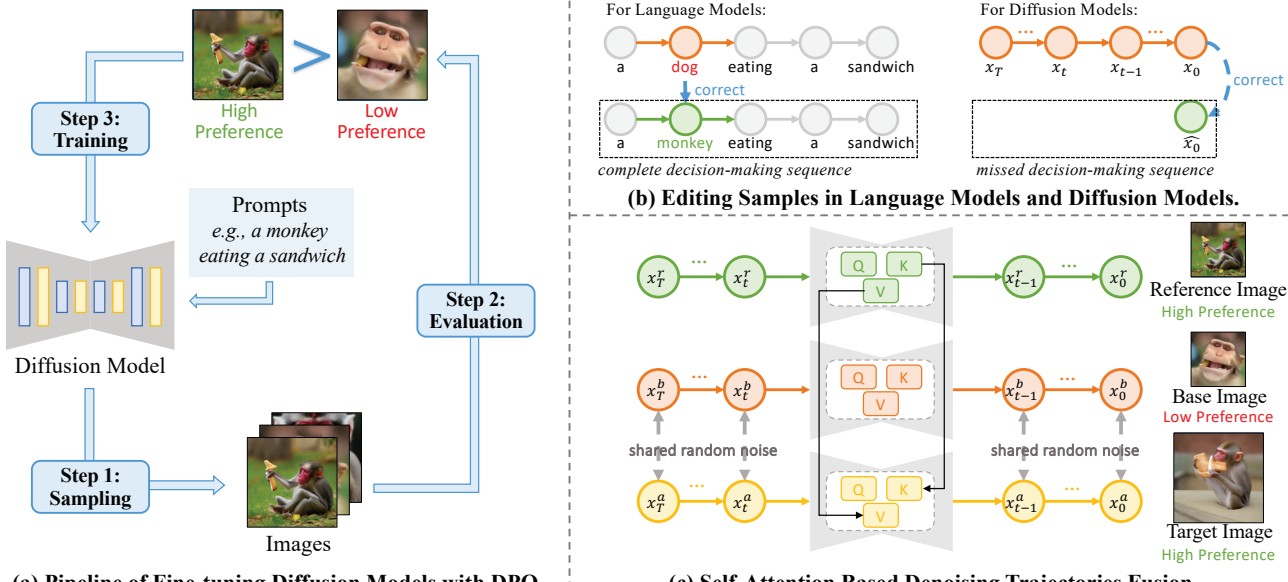

(a) Pipeline of Fine-tuning Diffusion Models with DPO.

(b) Editing Samples in Language Models and Diffusion Models.

(c) Self-Attention Based Denoising Trajectories Fusion.

*Figure 2.* (**Visual Inconsistency**) When people train diffusion models with direct preference optimization (DPO), the visual disparity between well-aligned and poorly-aligned images are enormous. This visual inconsistency limits the success of DPO in enhancing diffusion models. Meanwhile, the visually consistent samples obtained through manual editing lack denoising trajectories and are not suitable for RL training. To this end, we introduce D-Fusion, which constructs RL-trainable visually consistent samples.

prompts, and then evaluate the alignment between images and prompts via human preference or model prediction. These sampled images, along with their preference orders and denoising trajectories, can be further used in DPO to enhance the alignment of diffusion model.

However, DPO has so far achieved limited success in improving prompt-image alignment, primarily due to the issue of ***visual inconsistency*** in the training data. Visual inconsistency refers to the disparity between images in terms of structure, style or appearance, which is commonly observed in the images denoised from different noises. As shown in Figure 2(a), high-preference images (*i.e.*, well-aligned images) differ from low-preference images (*i.e.*, poorly-aligned images) not only in the alignment-related factors, but also in unrelated factors (*e.g.*, background). Interfered by unrelated factors, it is difficult for the model to identify which factors contribute positively to alignment. Meanwhile, with great differences, it is difficult to tell which image aligns better sometimes (*e.g.*, an image with only monkey and an image with only sandwich). As a result, learning effective denoising policies from those samples becomes challenging.

We believe that performing DPO with visually consistent image pairs can help diffusion models learn effective policies. Recent studies have corroborated similar perspectives on RL training of large language models (LLMs) and multimodal large language models (MLLMs) (Kong et al., 2025; Yu et al., 2024). As illustrated in Figure 2(b), these studies make fine-grained editing or annotations to the hallucina-

tions (Huang et al., 2024; Bai et al., 2024) present in the text output of the language model, thereby obtaining factual training data (*i.e.*, high-preference data) with consistent linguistic style. Since the editing or annotations to text output are at the ***token-level***, and the decision-making sequence in language model is constructed ***token-by-token*** (Rafailov et al., 2024), RL training can still proceed. By performing DPO with the pairs consisting of hallucinated text and corresponding factual text, the language model receives dense *reward* signals and achieves fine-grained alignment.

However, these methods on language models fail when applied to the text-to-image diffusion models. As illustrated in Figure 2(b), the decision-making sequence of the diffusion model is constructed ***timestep-by-timestep***. Existing editing methods, such as manual editing, Imagen Editor (Wang et al., 2023) and Imagic (Kawar et al., 2023), are capable of both aligning images and maintaining visual consistency. Yet, these methods perform editing at the ***pixel-level***, causing the loss of the timestep-by-timestep decision-making sequences. Once edited to better align with the prompts, these images *lack corresponding denoising trajectories*, making them unsuitable for RL fine-tuning. This motivates us to ask: *How can we generate RL/DPO-trainable visually consistent image pairs to fine-tune diffusion models?*

In this paper, we address this challenge by introducing D-Fusion: self-attention based **D**enoising trajectory **Fusion**, a method to construct RL-trainable visually consistent images. Our method offers key innovations in two phases. (1) In the

sampling phase, we propose to apply self-attention fusion between a high-preference sample (called reference image) and a low-preference sample (called base image) under the guidance of an auto-extracted mask to obtain a new sample (called target image), as illustrated in Figure 2(c). The mask, which is derived from the denoising process of the reference image, can reveal the position of alignment-related area. By applying self-attention fusion in the alignment-related area, the target image becomes as well-aligned as the reference image. Simultaneously, with shared random noise, the target image exhibits a high degree of visual consistency with the base image. (2) In the training phase, since the self-attention fusion is applied step-by-step along the denoising process, we collect the intermediate states to form the trajectories, which are the necessity of RL training. By performing DPO between the base images and corresponding target images, the diffusion models can achieve better prompt-image alignment than those fine-tuned with naive samples.

We conduct comprehensive experiments with three lists of prompts on Stable Diffusion (Rombach et al., 2022). The three prompt lists respectively consider the behavior of the object, the attribute of the object, and the positional relationship between the objects, which we believe can encompass a broad spectrum of commonly used prompt types in image generation. Furthermore, we apply D-Fusion to a variety of RL algorithms for fine-tuning diffusion models, including DPO (Wallace et al., 2023), DDPO (Black et al., 2024) and DPOK (Fan et al., 2023). Experimental results show that D-Fusion can effectively enhance the alignment of diffusion models across different prompts, and is compatible with different RL algorithms.

The main contribution of this work can be summarized as [1]: (1) We for the first time highlight the necessity of fine-tuning diffusion models with visually consistent image pairs when applying DPO, and discuss the challenge in obtaining RL-trainable visually consistent images. (2) We introduce D-Fusion, a compatible approach to construct visually consistent samples and corresponding denoising trajectories, where the latter is curial for RL training, to address the above challenge. (3) Comprehensive experimental results demonstrate the effectiveness of D-Fusion in improving prompt-image alignment when applied to different prompts and different RL algorithms.

## 2. Related Work

### 2.1. Controllable Generation with Diffusion Models

Diffusion models have demonstrated impressive ability in generating high-quality and high-fidelity images (Ho et al., 2020; Song & Ermon, 2020; Peebles & Xie, 2023). With

---

[1]The code for this work is available at this repository: https://github.com/hu-zijing/D-Fusion.

the increasing demand for more interactive and user-driven generation, researchers begin exploring methods to incorporate controllability into these models (Cao et al., 2024; Tong et al., 2023). A variety of studies aim to control the generation process of diffusion models with specific conditions, such as class labels (Dhariwal & Nichol, 2021; Ho & Salimans, 2022), layouts (Zheng et al., 2024), images (Preechakul et al., 2022) and audios (Yang et al., 2023). With the introduction of text encoders, diffusion models gain the ability to generate images from text (Rombach et al., 2022). Subsequent studies therefore focus on fine-tuning the pre-trained text-to-image diffusion models to improving alignment (Jiang et al., 2024; Lee et al., 2023). Among them, RL has been widely employed to enhance the controllability of diffusion models (Black et al., 2024; Clark et al., 2024; Fan et al., 2023; Wallace et al., 2023; Xu et al., 2023; Yang et al., 2024a;b; Hu et al., 2025). In this paper, by mitigating the issue of visual inconsistency, we further improve the performance of RL in training diffusion models.

### 2.2. Reinforcement Learning with Fine-grained Data

Reinforcement learning is a training paradigm that has played an important role in improving alignment of both diffusion models and language models. In the area of trustworthy LLMs/MLLMs, alignment to human preference has attracted widespread attention (Liu et al., 2024; Tu et al., 2023; Zhu et al., 2025; Yang et al., 2025), where reinforcement learning from human feedback (RLHF) has been employed accordingly (Bai et al., 2022; Rafailov et al., 2024; Ouyang et al., 2022). Current language models generate text in an auto-regressive manner (Vaswani et al., 2023), thus the token-by-token generation process can be regarded as a Markov decision process. Recently, researchers perform fine-grained corrections or assign fine-grained human feedback to the textual training data (Kong et al., 2025; Yu et al., 2024; Wu et al., 2023). Fine-grained data can provide dense reward signals to RL, thus achieving impressive fine-tuning results. In this paper, by employing denoising trajectory fusion, we provide visually consistent samples for RL training of diffusion models, which have similar fine-grained effects.

### 2.3. Attention Control for Diffusion Models

The attention mechanism has garnered considerable interest and sparked a wealth of research (Vaswani et al., 2023; Dosovitskiy et al., 2021; Wang et al., 2024). In diffusion models, some studies have demonstrated how cross-attention maps in the denoising process determine the layouts of generated images (Hertz et al., 2022; Brooks et al., 2023; Mokady et al., 2022). Additionally, other studies have explored the role of self-attention layers in these models (Cao et al., 2023; Tumanyan et al., 2022; Shi et al., 2024). These studies can edit images by controlling the attention layers, and some of them have the potential to preserve the denoising tra-

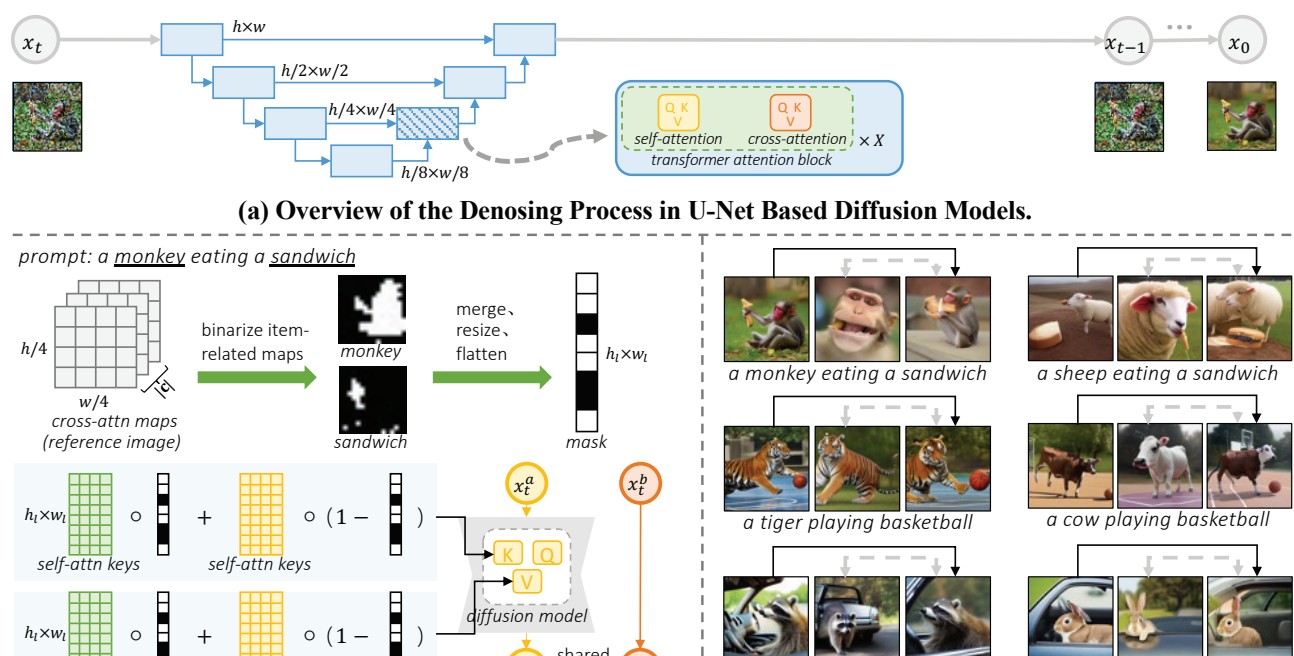

**(a) Overview of the Denosing Process in U-Net Based Diffusion Models.**

**(b) Denoising Trajectory Fusion.**

**(c) Visually Consistent Samples.**

*Figure 3.* (**Method Overview**) We propose D-Fusion to construct RL-trainable visually consistent samples. (a) Each layer of the U-Net based diffusion models contains several transformer attention blocks, and each block contains a self-attention module and a cross-attention module. (b) D-Fusion constructs visually consistent samples through two steps: cross-attention mask extraction and self-attention fusion. (c) Examples of visually consistent samples. Each set consists of three images: the reference image, the base image, and the target image. The target images are not only as well-aligned as the reference images but also maintain visual consistency with the base images.

jectories. However, these methods generally transfer an image from one prompt to another, which does not correspond to our task of aligning an image with corresponding prompt. Nevertheless, these methods inspire us to explore attention-based techniques in this paper. We present some observations on these methods in Appendix B.

## 3. Method

In this section, we start by formulating the problem, followed by a detailed introduction to D-Fusion, covering both the sampling and training phases.

### 3.1. Problem Formulation

**Text-to-Image Diffusion Models.** In this work, we consider pre-trained text-to-image diffusion models $p(\mathbf{x}_0 \mid \mathbf{c})$, which generate a sample $\mathbf{x}_0$ conditioned on a textual prompt $\mathbf{c}$. Beginning with random noise $\mathbf{x}_T \sim \mathcal{N}(\mathbf{0}, \mathbf{I})$, diffusion models iteratively transform the noise through $T$ steps into a clear image $\mathbf{x}_0$ that matches the given prompt (Sohl-Dickstein et al., 2015; Dhariwal & Nichol, 2021). Building upon the samplers of DDPM (Ho et al., 2020) and DDIM (Song et al.,

2022), each iteration is performed by applying the following denoising formula:

$$p_\theta(\mathbf{x}_{t-1} \mid \mathbf{x}_t, \mathbf{c}) = \mathcal{N}(\mathbf{x}_{t-1} \mid \mu_\theta(\mathbf{x}_t, t, \mathbf{c}), \sigma_t \mathbf{I}^2), \quad (1)$$

where $t$ denotes current timestep, $\mu_\theta$ represents the prediction made by a diffusion model parameterized by $\theta$, and $\sigma_t$ is the fixed timestep-dependent variance. The reverse process produces a denoising trajectory $\{\mathbf{x}_T, \mathbf{x}_{T-1}, \ldots, \mathbf{x}_0\}$ ending with a sample $\mathbf{x}_0$.

**Attention Mechanism in Diffusion Models.** Transformer attention blocks (Vaswani et al., 2023) have been applied in each layer of the U-Net (Ronneberger et al., 2015) based diffusion models. As shown in Figure 3(a), the U-Net based diffusion models contain several down-sampling layers, a middle layer, and corresponding up-sampling layers. Furthermore, each layer contains several transformer attention blocks, and each attention block in diffusion models contains a self-attention module and a cross-attention module. The attention mechanism can be formulated as follows:

$$\text{Attention}(Q, K, V) = \text{softmax}(\frac{QK^T}{\sqrt{d_{key}}})V, \quad (2)$$

where $Q \in \mathbb{R}^{m \times d_{key}}$ are queries projected from image features, and $K \in \mathbb{R}^{n \times d_{key}}$, $V \in \mathbb{R}^{n \times d_{value}}$ are keys and values projected from image features (in self-attention module) or prompt embeddings (in cross-attention module). In this formula, $\text{softmax}(\frac{QK^T}{\sqrt{d_{key}}})$ is commonly referred to as attention maps, represented by $A$.

**Denoising as a Decision-Making Problem.** The denoising process in diffusion models can be formulated as a *sequential decision-making problem*. The process can be defined by a tuple $(\mathcal{S}, \mathcal{A}, P, R, \pi_\theta)$, where $\mathcal{S}$ is the state space, $\mathcal{A}$ is the action space, $P$ is the transition function, $R$ is the reward function, and $\pi_\theta$ is the decision-making policy. At each timestep $t$, the *state* $s_t \in \mathcal{S}$ is represented by $(\mathbf{c}, t, \mathbf{x}_t)$, *i.e.*, the text prompt, the current timestep, and the noisy image at the current timestep. The *action* $a_t \in \mathcal{A}$ refers to the denoising operation that generates the next noisy image $\mathbf{x}_{t-1}$. The *transition* $P(s_{t+1} \mid s_t, a_t)$ specifies the distribution over the next state $s_{t+1}$ given the current state $s_t$ and action $a_t$, and is provided by corresponding samplers in DDPM and DDIM. The *reward* $R(\mathbf{c}, \mathbf{x}_0)$ corresponds to the prompt-image alignment score in our settings, which can be given by human preference or model evaluation. And the *policy* is defined as $\pi_\theta(a_t \mid s_t) = p_\theta(\mathbf{x}_{t-1} \mid \mathbf{x}_t, \mathbf{c})$, which describes how to select the current action based on current state. By adopting this formulation, we can enhance the prompt-image alignment in diffusion models by maximizing the following objective:

$$\mathcal{J}_{\text{RL}}(\theta) = \mathbb{E}_{\mathbf{c} \sim p(\mathbf{c}), \mathbf{x}_0 \sim p_\theta(\mathbf{x}_0 \mid \mathbf{c})} \left[ R(\mathbf{x}_0, \mathbf{c}) \right], \qquad (3)$$

where $p(\mathbf{c})$ is a uniform distribution over the candidate prompts.

### 3.2. Sampling: Denoising Trajectory Fusion

When employing reinforcement learning, people first sample a set of images $I_1, \ldots, I_n$, and reserve their denoising trajectories. These images contain both well-aligned and poorly-aligned ones, and can be further used as training data for RL. We refer to these well-aligned images as reference images and poorly aligned-images as base images. The goal of our method is to generate a target image $I^a$ that is both as well-aligned as a given reference image $I^r$, and visually consistent with a given base image $I^b$, where $I^r$ and $I^b$ are generated with the same textual prompt $\mathbf{c}$. D-Fusion reaches this goal through the following two steps: cross-attention mask extraction and self-attention fusion. Formally, the denoising trajectories of $I^r$ and $I^b$ are represented as $\{\mathbf{x}^r_T, \mathbf{x}^r_{T-1}, \ldots, \mathbf{x}^r_0\}$ and $\{\mathbf{x}^b_T, \mathbf{x}^b_{T-1}, \ldots, \mathbf{x}^b_0\}$ respectively.

**Cross-Attention Mask Extraction.** Firstly, we extract a mask $M_t$ from reference image at each timestep $t$, *i.e.*, at the denoising process from $\mathbf{x}^r_t$ to $\mathbf{x}^r_{t-1}$. Let $h \times w$ represent the image resolution of $\mathbf{x}^r_t$ ($h \times w = 64 \times 64$ in Stable

Diffusion), and $h_l \times w_l$ represent the image resolution at layer $l$ of the U-Net. Inspired by previous work (Hertz et al., 2022; Cao et al., 2023), the cross-attention maps contain sufficient information about shapes and structures of the generated images, among which the first up-sampling layer (with resolution $\frac{h}{4} \times \frac{w}{4} = 16 \times 16$ in Stable Diffusion) performs the best. Therefore, we average the cross-attention maps across all heads and all attention blocks in the first up-sampling layer, and extract a mask from them.

Formally, after averaging and reshaping, the cross-attention maps are denoted as $A^{cross}_t \in \mathbb{R}^{\frac{h}{4} \times \frac{w}{4} \times |\mathbf{c}|}$, where $|\mathbf{c}|$ is the number of tokens in prompt $\mathbf{c}$. The $i$-th attention map $A^{cross}_t[:, :, i]$ indicates the extent to which each pixel in the image should pay attention to the $i$-th token in the prompt. Let $\mathcal{O}_{\mathbf{c}} = \{o_1, \ldots, o_k\}$ represents index list of the item-related tokens in prompt $\mathbf{c}$, then the mask $M_t$ can be extracted with the following formula:

$$M_t = \bigoplus_{o \sim \mathcal{O}_{\mathbf{c}}} \left[ (A^{cross}_t[:, :, o] \geq thr_o)?\mathbf{1} : \mathbf{0} \right], \qquad (4)$$

where $thr_o$ is a hyperparameter that defines the mask threshold for the $o$-th token, $\mathbf{1}$ is the all-one matrix, and $\mathbf{0}$ is the all-zero matrix. In this formula, we first binarize the attention maps to generate the masks for corresponding items, as shown in the Figure 3(b). Afterwards, we merge them into one mask through XOR operation. The resulting mask $M_t \in \mathbb{B}^{\frac{h}{4} \times \frac{w}{4}}$, where $\mathbb{B} = \{0, 1\}$, reveals the position of the items mentioned by the prompt in the reference image.

**Self-Attention Fusion.** To generate an ideal target image $I^a$, our approach is based on the idea of having the prompt-related area imitate the reference image $I^r$, while the other area retain the features of base image $I^b$. Inspired by previous work (Cao et al., 2023; Tumanyan et al., 2022), the $(i, j)$ entry in the self-attention maps $A^{self}_t \in \mathbb{R}^{(h_l \times w_l) \times (h_l \times w_l)}$ indicates the extent to which the $i$-th pixel in the image should pay attention to the $j$-th pixel at timestep $t$. Therefore, we design the mechanism named self-attention fusion, to control the attention allocation in $I^a$. Starting with the same random noise as $I^b$, we progressively denoise it with diffusion model, and apply self-attention fusion at each timestep $t$ as follows. Firstly, we resize the mask $M_t$ to match the image resolution of the current layer, resulting in a new mask $\widehat{M_t} \in \mathbb{B}^{h_l \times w_l}$. Afterwards, we manipulate the keys and values in self-attention as Eq.(5):

$$K^a_{new} = K^r \circ \text{Flatten}(\widehat{M_t}) + K^a \circ \left( \mathbf{1} - \text{Flatten}(\widehat{M_t}) \right),$$
$$V^a_{new} = V^r \circ \text{Flatten}(\widehat{M_t}) + V^a \circ \left( \mathbf{1} - \text{Flatten}(\widehat{M_t}) \right),$$
$$(5)$$

where the signal $\circ$ is Hadamard product [2], the $K^r \in \mathbb{R}^{(h_l \times w_l) \times d_{key}}, V^r \in \mathbb{R}^{(h_l \times w_l) \times d_{value}}$ are keys and val-

---

[2] For two matrices $A$ and $B$, the Hadamard product is $A \circ B = [a_{ij}] \circ [b_{ij}] = [a_{ij}b_{ij}]$.

ues of $I^r$, the $K^a, V^a$ are keys and values of $I^a$ generated at current denoising step, and Flatten() reshapes $\widehat{M}_t$ into $\mathbb{B}^{(h_l \times w_l) \times 1}$, enabling it to compute the Hadamard product with the keys and values after auto-broadcasting.

By applying self-attention fusion, the diffusion model can generate the target image $I^a$ that is not only as well-aligned as $I^r$, but also visually consistent with $I^b$. On one hand, by injecting the fused keys $K^a_{new}$, the self-attention maps allocate attention to the prompt-related area with reference to $I^r$. Subsequently, by injecting the fused values $V^a_{new}$, the final image features in the prompt-related area also take into account the features of $I^r$. Thus the prompt-related area in $I^a$ becomes well-aligned, as the reference image $I^r$ goes. On the other hand, by sharing the same random noise with $I^b$, retaining the original queries, and retaining the keys and values in prompt-unrelated area, the target image $I^a$ also achieves visual consistency with $I^b$.

### 3.3. Training: DPO with Visually Consistent Samples

By applying denoising trajectory fusion based on $I^b$ and with reference to $I^r$, we can get the well-aligned target image $I^a$ along with its denoising trajectory $\{\mathbf{x}^a_T, \mathbf{x}^a_{T-1}, \dots, \mathbf{x}^a_0\}$. The direct preference optimization can therefore be employed with the image pair consisting of high-preference image $I^a$ and low-preference image $I^b$. Following traditional DPO (Wallace et al., 2023; Yang et al., 2024a), the diffusion model with parameters $\theta$ can be optimized with following objective:

$$-\mathbb{E}\Big(\sum_{t=1}^{T} \log \sigma \Big[\beta \frac{p_\theta(\mathbf{x}^a_{t-1} \mid \mathbf{x}^a_t, \mathbf{c})}{p_{\theta_{\text{old}}}(\mathbf{x}^a_{t-1} \mid \mathbf{x}^a_t, \mathbf{c})} - \beta \frac{p_\theta(\mathbf{x}^b_{t-1} \mid \mathbf{x}^b_t, \mathbf{c})}{p_{\theta_{\text{old}}}(\mathbf{x}^b_{t-1} \mid \mathbf{x}^b_t, \mathbf{c})}\Big]\Big),$$
(6)

where $\sigma$ is the sigmoid function, $\theta_{\text{old}}$ is the parameters of diffusion model prior to update, and $\beta$ is a hyperparameter controlling the deviation from $p_\theta$ to $p_{\theta_{\text{old}}}$. By introducing CLIP (Radford et al., 2021) to replace human in evaluating prompt-image alignment, it becomes possible to conduct multiple rounds of online learning, allowing the model to progressively adapt to a new image distribution that aligns well with corresponding prompts. Beyond DPO, we further employ DDPO (Black et al., 2024) and DPOK (Fan et al., 2023) to fine-tune diffusion models with visually consistent samples. For a comprehensive description of implementation details, we refer the readers to Appendix C.

## 4. Experiments

In this section, we demonstrate the effectiveness of D-Fusion both qualitatively and quantitatively. Afterwards, we focus on ablation studies on denoising trajectories and RL algorithms, as well as demonstrating the generalization ability. For simplicity, we refer to DPO+D-Fusion (*i.e.*, employing DPO with D-Fusion) as our method in some places.

### 4.1. Experimental Setup

**Diffusion Models.** We use Stable Diffusion (SD) 2.1-base (Rombach et al., 2022), one of the most advanced diffusion models, as the base model for the experiments. We employ DDIM (Song et al., 2022) as the sampler. The weight of noise in DDIM sampler is set to 1.0, which decides the degree of randomness at each denoising step. We apply Low-Rank Adaptation (LoRA) (Hu et al., 2021) for efficient fine-tuning. Following the previous work (Black et al., 2024), the total denoising timesteps $T$ is set to 20. Each experiment is conducted with three different seeds.

**Prompt Templates.** We construct the prompt lists based on three templates. The three templates consider the behavior of the object, the attribute of the object, and the positional relationship between the objects respectively. (1) Template 1: "*a(n) [animal] [activity]*". The animal is chosen from the the list of 45 common animals given by previous work (Black et al., 2024), and the activity is chosen from the list: "*eating a sandwich*", "*driving a car*" and "*playing basketball*". (2) Template 2: "*a(n) [color] and [material] [object]*". We select six common colors (*e.g.*, red) and nine common materials (*e.g.*, wooden) for this template. The object list is chosen from the Visual Relation Dataset (VRD) (Lu et al., 2016). We randomly combine colors, materials, and objects to form the prompts. (3) Template 3: "*the [object 1] [predicate] the [object 2]*". We select four position-related predicates: "*above*", "*below*", "*on the left of*" and "*on the right of*". We construct the prompts based on the annotations of VRD to ensure their rationality. The prompt list for each template contains 40 prompts for training, and 40 prompts for generalization test. We present the full prompt lists in Appendix G.

**Rewards and Evaluation Metrics.** We evaluate the prompt-image alignment by CLIPScore (Hessel et al., 2022), and also use it as the reward function (if needed). A higher CLIPScore represents better alignment. In terms of implementation, we use Vit-H-14 CLIP model (Radford et al., 2021; Ilharco et al., 2021).

### 4.2. Qualitative Evaluation

We first evaluate the performance of D-Fusion when applied to DPO (Wallace et al., 2023; Yang et al., 2024a). After employing DPO with or without D-Fusion for the same training rounds, we sample a series of images from original model and fine-tuned models with same random seeds, as shown in Figure 4(top). The results qualitatively show that training diffusion models with visually consistent samples yields better performance in improving prompt-image alignment than training without them when employing DPO. We also conduct a human preference test with 22 independent human raters (ranging from undergraduates to Ph.D.), who are asked to select the image that best aligns with corresponding

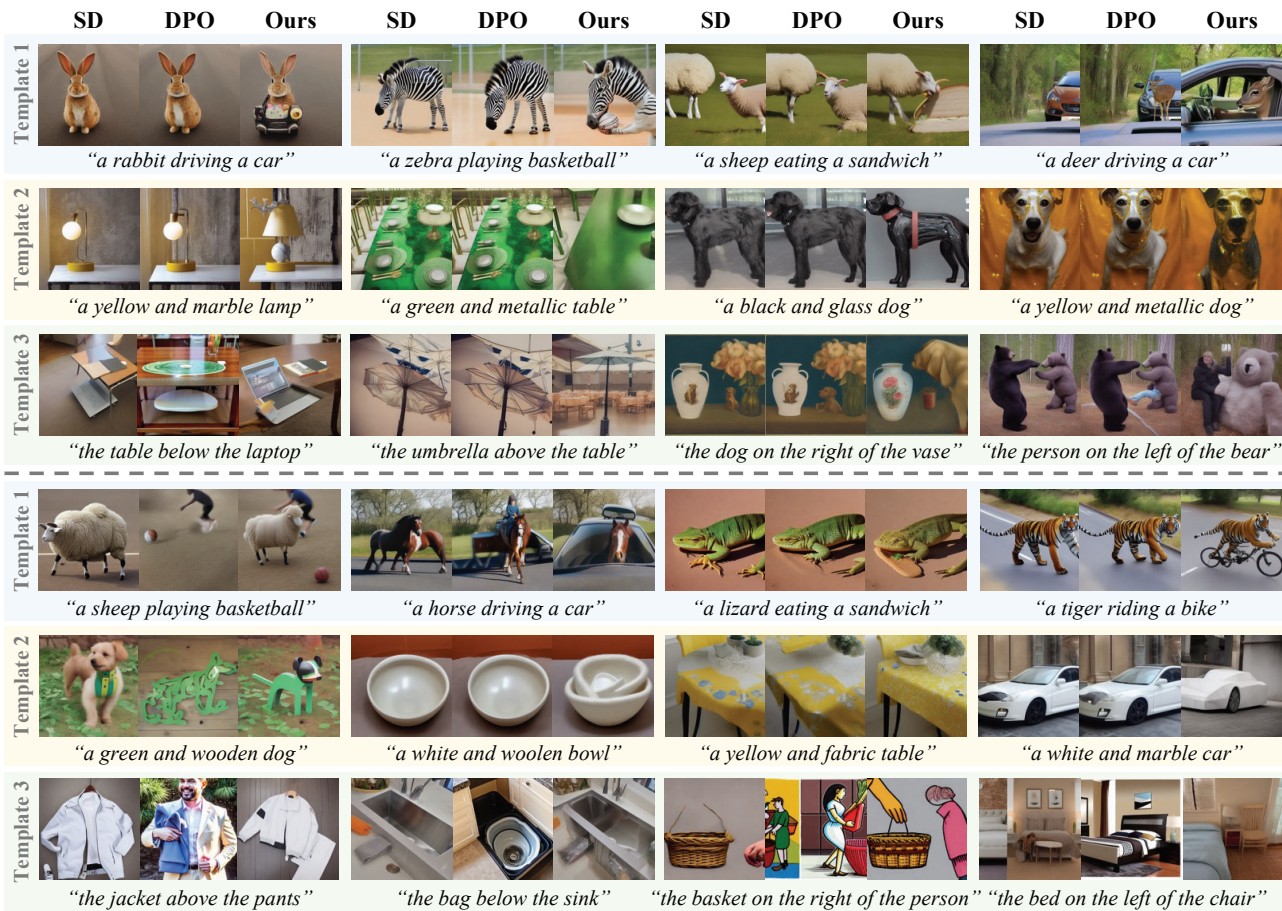

*Figure 4.* (**Qualitative Results**) Examples of images generated by original model and fine-tuned models on three templates. For each set of images, we use the same random seed. For both training prompts (top) and test prompts (bottom), the models fine-tuned by DPO+D-Fusion achieves better prompt-image alignment compared to the original model and the models fine-tuned by naive DPO.

prompt from a set of three images generated by different models. We report the average preference rates in Figure 6. The results indicate that the preference rates of the images generated by the models fine-tuned with our method consistently outperform those by original model (SD) and by the models fine-tuned with naive DPO on the three prompt templates. We present more samples in Appendix F.

### 4.3. Quantitative Evaluation

We also quantitatively demonstrate the alignment of the models fine-tuned by DPO with or without D-Fusion. As shown in Figure 5, we conduct multiple rounds of fine-tuning on the diffusion models with different methods. At each round, we use the same seed to sample the fine-tuned different models, and test the alignment scores of the generated images. The results illustrate the alignment scores as the training progresses on the three prompt templates, where the x-axis represents the amount of image data used to fine-tune the models, and y-axis represents the CLIPScore. It can be seen that after training with the same amount of data,

the models fine-tuned by our method almost always achieve higher alignment scores than the models fine-tuned by naive DPO. These results indicate that training with visually consistent samples can enhance diffusion models to a greater extent than training with naive DPO.

### 4.4. Ablation Study

We conduct ablation studies on denoising trajectories and RL algorithms. For the former, we investigate the effectiveness of training with the denoising trajectories generated through DDIM inversion (Song et al., 2022; Mokady et al., 2022). For the latter, we apply D-Fusion across different RL algorithms to assess its compatibility and performance.

**Comparison with DDIM Inversion.** The goal of DDIM inversion is to estimate the initial random noise $\mathbf{x}_T^{inv}$ from a clear image $\mathbf{x}_0$ step by step, thus constructing a denoising trajectory $\{\mathbf{x}_0, \mathbf{x}_1^{inv}, \ldots, \mathbf{x}_T^{inv}\}$ in reverse order. In this ablation study, we apply DDIM inversion to visually consistent samples, and utilize the generated denoising trajec-

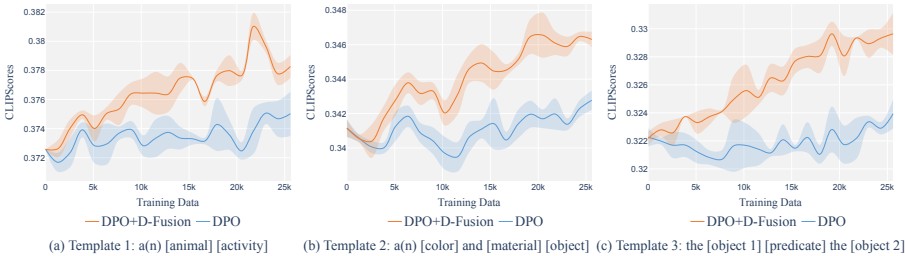

(a) Template 1: a(n) [animal] [activity]  (b) Template 2: a(n) [color] and [material] [object]  (c) Template 3: the [object 1] [predicate] the [object 2]

*Figure 5.* (**Alignment**) Alignment curves of the diffusion models fine-tuned with or without D-Fusion on three prompt templates. Results show that training with D-Fusion can enhance the alignment of diffusion models to a greater extent.

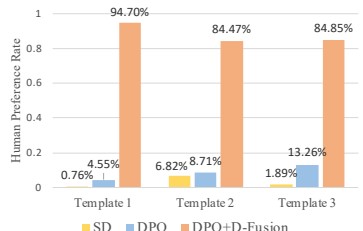

*Figure 6.* (**Human Evaluation**) Human preference rates for prompt-image alignment of the images generated by SD, DPO and our method.

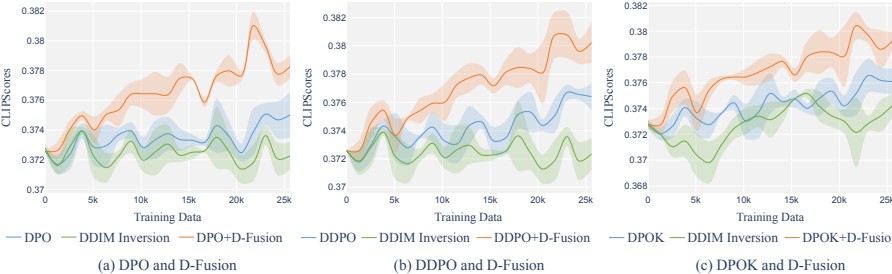

(a) DPO and D-Fusion  (b) DDPO and D-Fusion  (c) DPOK and D-Fusion

*Figure 7.* (**Ablation Study**) The ablation studies on denoising trajectories and RL algorithms with template 1. Results indicate that (1) Constructing denoising trajectories by DDIM inversion is not a practical way; (2) Integrating D-Fusion can enhance the effect of different RL algorithms.

*Table 1.* (**Generalization**) Prompt-image alignment (measured by CLIPScore ↑) of the images generated by the SD, DPO, and our method on three templates. The prompts for generalization test are not used during training.

| Methods | Temp.1 | Temp.2 | Temp.3 |
|---------|--------|--------|--------|
| SD | 0.3725 | 0.3396 | 0.3213 |
| DPO | 0.3733 | 0.3407 | 0.3230 |
| Ours | **0.3758** | **0.3446** | **0.3276** |

tories to replace those provided by D-Fusion, which are then used to train the models. As shown in Figure 7, the results reveal that after applying DDIM inversion, the models do not receive any noticeable improvement in alignment. This indicates that DDIM inversion fails to provide accurate noise estimations. Previous work (Mokady et al., 2022) has noted that, if denoising from $\mathbf{x}_T^{inv}$ to reconstruct the image, the reconstructed one exhibited visible difference from the original one, which is a consistent phenomenon with our observation here. The reason is DDIM inversion relies on a rough assumption that $\epsilon_\theta(\mathbf{x}_{t-1}, t) = \epsilon_\theta(\mathbf{x}_t, t)$, leading to inaccurate estimations. Therefore, compared to our method, DDIM inversion is not a practical approach to construct denoising trajectories for RL training.

**Compatibility with Different RL Algorithms.** D-Fusion demonstrates compatibility with a variety of RL algorithms. In this ablation study, we further apply D-Fusion to the widely used RL-based diffusion fine-tuning methods, including DDPO (Black et al., 2024) and DPOK (Fan et al., 2023). The implementation details are presented in Appendix C. As shown in Figure 7, among these methods, the integration of D-Fusion enhances the alignment of diffusion models to a greater extent. More experimental results on template 2 and 3 are shown in Appendix E. The results demonstrate that training with visually consistent samples is effective across different RL algorithms.

### 4.5. Generalization Ability

The models fine-tuned with our method exhibit generalization capabilities, further enhancing the potential for real-world applications. As shown in Table 1, for each prompt template, we use different models to separately sample 1,280 images with the same random seeds. The prompts used here are not optimized with RL fine-tuning, but accord with corresponding template. The results indicate that the images generated by the models fine-tuned by our method achieve higher alignment scores compared to those generated by original model (SD) and DPO. Figure 4(bottom) presents the image examples generated with these prompts, qualitatively showing generalization ability of the models fine-tuned with our method. For more image examples, we refer the readers to Appendix F.

## 5. Conclusion

In this work, we mitigate the issue of prompt-image misalignment in diffusion models by employing direct preference optimization with visually consistent samples. We highlight the challenge of obtaining RL-trainable visually consistent samples. To address this challenge, we introduce D-Fusion, a self-attention based method that can not only generates visually consistent and well-aligned samples from given images, but also retain the denoising trajectories. We

conduct comprehensive experiments using Stable Diffusion as backbone, incorporating a variety of prompts and RL algorithms. Both qualitative and quantitative experimental results demonstrate that, by training with visually consistent samples generated by D-Fusion, the RL-based fine-tuning can achieve better prompt-image alignment.

## Acknowledgements

This work was supported in part by the National Key Research and Development Program of China (2024YFE0203700), National Natural Science Foundation of China (62376243, 62441605), and "Pioneer" and "Leading Goose" R&D Program of Zhejiang (2025C02037). All opinions in this paper are those of the authors and donot necessarily reflect the views of the funding agencies.

## Impact Statement

This paper presents work whose goal is to advance the field of Machine Learning. There are many potential societal consequences of our work, none which we feel must be specifically highlighted here.

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

The **Appendix** is organized as follows:

## A. Abbreviation and Symbol Table

The abbreviations and symbols used in this paper are presented in Table 2.

## B. Observations on Attention Control

In this section, we present some observations on attention control from three perspectives: (1) What are the effects of different attention control methods (*i.e.*, fusing different components in the attention module). (2) How do the timesteps and layers in U-Net affect the fusion results. (3) How do the cross-attention maps look like. We use prompt "*a cat playing chess*" in these observations.

(1) ***What are the effects of different attention control methods?*** We have observed varieties of different attention control methods. As shown in Figure 8(top), subfigures (a) to (c) correspond to the previous work (Tumanyan et al., 2022; Cao et al., 2023; Hertz et al., 2022), where they inject the self-attention maps, the keys and values of self-attention, and the cross-attention maps from the reference image into the base image, respectively. Subfigure (a) illustrates that injecting self-attention maps can easily lead to significant blurring in the resulting image. Subfigures (b) and (c) inject the keys and values of self-attention, and the cross-attention maps, respectively. In the former, the features largely mimic those of the reference image, while in the latter, the features from the base image are better preserved. For instance, the table in subfigure (c) appears green, just like in base image, whereas subfigure (b) does not exhibit this characteristic. Meanwhile, we experiment with injecting additional components in attention mechanism, as shown in subfigures (d) to (f). The images in Figure 8(top) are either blurred or retain too few features from the base image.

In order to generate ideal images, we introduce masks as in the previous work MasaCtrl (Cao et al., 2023). MasaCtrl

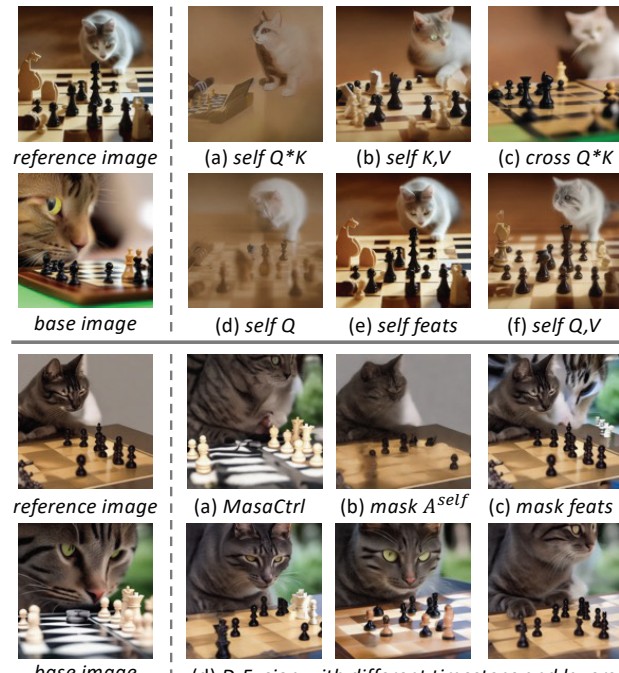

**Figure 8.** Effects of different attention control methods.

applies masks to two components: self-attention maps and image features (*i.e.*, the output of the attention module). The masks are used to make the foreground of resulting images resemble the reference images, while the background resembles the base images. As shown in subfigures (a) to (c) of Figure 8(bottom), we test MasaCtrl and its two parts respectively. They fail to generate ideal images primarily because the image features are directly tied to the pixel structure. Therefore, applying masks to image features often leads to confusion at the boundaries between the covered and uncovered areas. We can conclude that MasaCtrl is more suitable for fusing two images with similar foreground and background boundaries, such as the images generated with same seed (but different prompts). Therefore, we turn to apply masks to keys and values in D-Fusion, and get robust fusion effects, as shown in subfigure (d) of Figure 8(bottom).

(2) ***How do the timesteps and layers in U-Net affect the fusion results?*** The U-Net in Stable Diffusion has three down-sampling layers, one middle layer and three up-sampling layers. We number them in order from 0 to 7. As shown in Figure 9, we apply D-Fusion at different timesteps and different U-Net layers. The x-axis represents fused timesteps, and the y-axis represents fused layers. We present some of the most representative layers, specifically layer 3 (*i.e.*, middle layer), layers 2-4 (*i.e.*, middle layer, the last down-sampling layer and the first up-sampling layer), layers 3-6 (*i.e.*, the whole up-sampling layers) and layers 3-5 (*i.e.*, the whole up-sampling layers except for the last one).

*Table 2.* List of important abbreviations and symbols.

| Abbreviation/Symbol | Meaning |
| --- | --- |
| *Abbreviations of Concepts* | |
| DM | Diffusion Model |
| RL | Reinforcement Learning |
| DPO | Direct Preference Optimization |
| SD | Stable Diffusion |
| LoRA | Low-Rank Adaptation |
| DDIM | Denoising Diffusion Implicit Model |
| DDPM | Denoising Diffusion Probabilistic Model |
| CLIP | Contrastive Language-Image Pre-Training |
| *Abbreviations of Methods* | |
| D-Fusion | Self-attention based Denoising trajectory Fusion |
| DDPO | Denoising Diffusion Policy Optimization |
| DPOK | Diffusion Policy Optimization with KL regularization |
| *Symbols in Diffusion Models* | |
| $\mathbf{x}_0$ | Generated image |
| $\mathbf{x}_t$ | Noisy image at timestep $t$ |
| $\{\mathbf{x}_T, \mathbf{x}_{T-1}, \ldots, \mathbf{x}_0\}$ | Denoising trajectory |
| $\mathbf{c}$ | Condition for image generation, also called prompt |
| $\theta$ | Parameters of the diffusion model |
| $\mathcal{N}()$ | Gaussian distribution |
| $T$ | Total timesteps |
| *Symbols in Reinforcement Learning* | |
| $s_t$ | State at timestep $t$ |
| $a_t$ | Action at timestep $t$ |
| $\pi_\theta$ | Action selection policy parameterized by $\theta$ |
| $R$ | Reward function |
| $\hat{r}$ | Rewards after normalization |
| *Symbols in D-Fusion* | |
| $\mathbb{R}$ | Set of real numbers |
| $\mathbb{B}$ | Binary set $\{0, 1\}$ |
| $I^a, I^b, I^r$ | Target image [3], base image and reference image |
| $Q, K, V$ | Query, key and value in attention mechanism |
| $A_t^{cross}, A_t^{self}$ | Cross-attention maps and self-attention maps at timestep $t$ |
| $\mathcal{O}_{\mathbf{c}}$ | Index list of the item-related tokens in prompt $\mathbf{c}$ |
| $M_t$ | Cross-attention mask at timestep $t$ |
| $\circ$ | Hadamard product |
| $\oplus$ | Exclusive OR operation |

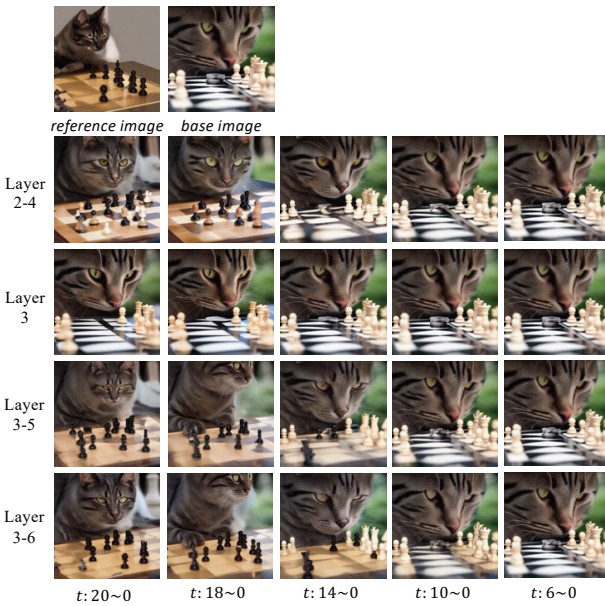

*Figure 9.* Impact of timesteps and U-Net layers on fusion results.

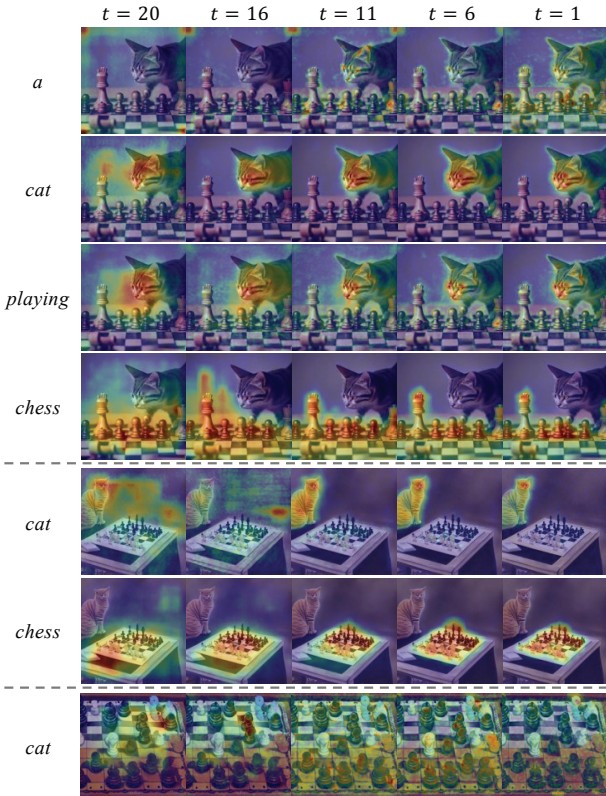

*Figure 10.* The heatmaps of cross-attention maps at different timesteps.

(3) ***How do the cross-attention maps look like?*** As shown in Figure 10, the cross-attention maps at each timestep are highly correlated with the corresponding items in the generated images. More specifically, in the early timesteps, the layouts of the images have not yet fully taken shape, so the cross-attention maps do not accurately identify the items' location. In the middle timesteps, the cross-attention maps gradually turn to mark the items' position precisely. By the later timesteps, the fundamental features of each item have been established, thus the cross-attention maps shift focus to more fine-grained details (*e.g.*, cat's face). For tokens that do not represent items, their cross-attention maps do not have an obvious meaning, but are generally highlighted in the areas of related items. If the image fails to align with the prompt, such as when there is no cat present, the corresponding cross-attention maps will also lack clearly highlighted areas.

## C. Implementation Details

### C.1. Detailed Implementation of Our Method

**Fusion Layers and Timesteps.** As shown in Appendix B, usually, employing self-attention fusion at all layers and all timesteps does not generate ideal images (*i.e.*, images that are both well-aligned and visually consistent with given poorly-aligned images). Therefore, we only employ self-attention fusion at some layers and some timesteps. For layers, we employ self-attention fusion at the middle layer and the up-sampling layers (*i.e.*, from layer 3 to layer 6 in Stable Diffusion 2.1-base). For timesteps, we employ self-attention fusion from timestep $t = 18$ to $t = 1$.

**Further Alignment Verification.** Although D-Fusion has demonstrated the ability to generate both well-aligned and visually consistent samples, it also generates failed cases sometimes. Excessive use of failed cases as training data can have a negative impact on fine-tuning the diffusion models. Therefore, we introduce additional verification before training, which is shown as follows:

$$R(\mathbf{x}_0^a, \mathbf{c}) - R(\mathbf{x}_0^b, \mathbf{c}) \geq thr_{ado} * \left( R(\mathbf{x}_0^r, \mathbf{c}) - R(\mathbf{x}_0^b, \mathbf{c}) \right),$$
(7)

where $thr_{ado}$ is a hyperparameter called adoption threshold (usually, $0.0 \leq thr_{ado} \leq 1.0$), and $R$ is reward function. If the target image $I^a$ does not meet the requirements, we will replace it with the reference image $I^r$. This replacement is reasonable, as the pairing between reference image $I^r$ and base image $I^b$ is consistent with that used in the naive DPO.

**Compatibility with DDPO.** Before optimization, the alignment scores evaluated by CLIP need to be normalized first. In implementation, we calculate the mean and standard deviation of the alignment scores for current and all the previous rounds. The scores from previous rounds are also used in calculation in order to increase the sample size under the

---

[3]We use $I^a$ instead of $I^t$ to represent the target image, as $t$ commonly refers to timestep in diffusion models.

*Table 3.* Hyperparameters of our experiments.

| | Hyperpatameter | D-Fusion | Baselines |
|---|---|---|---|
| Sampling | Denoising steps $T$ | 20 | 20 |
| | Noise weight $\eta$ | 1.0 | 1.0 |
| | Guidance scale | 5.0 | 5.0 |
| | Batch size | 4 | 4 |
| | Batch count | 160 | 160 |
| Fusion | Fusion U-Net layers | 3-6 | - |
| | Fusion timesteps | 18-1 | - |
| | Adoption threshold | 1.0 | - |
| Optimizer | Optimizer | AdamW | AdamW |
| | Learning rate | 1e-4 | 1e-4 |
| | Weight decay | 1e-4 | 1e-4 |
| | $(\beta_1, \beta_2)$ | (0.9, 0.999) | (0.9, 0.999) |
| | $\epsilon$ | 1e-8 | 1e-8 |
| | Grad. clip norm | 1.0 | 1.0 |
| Training | Batch size | 1 | 1 |
| | Grad. accum. steps | 320 | 320 |
| | Inner epoch | 2 | 2 |

## C.2. Experimental Resources

The experiments were conducted on 24GB NVIDIA 3090 and 4090 GPUs. It took approximately 30 hours to reach a training data volume of 25.6k when applying DPO and DDPO, and approximately 40 hours when applying DPOK.

## C.3. Hyperparameters

The hyperparameters of our experiments are listed in Table 3. Hyperparameters that are not listed keep consistent with the corresponding RL work (Wallace et al., 2023; Fan et al., 2023; Black et al., 2024).

# D. Pseudo-Code

The pseudo-code of employing direct preference optimization with D-Fusion for one training round is shown in Algorithm 1.

same prompt. With the mean and standard deviation of the image scores under the same prompt, we can normalize them by $\hat{r} = \frac{R(\mathbf{x}_0, \mathbf{c}) - mean(R(\mathbf{x}_0, \mathbf{c}))}{std(R(\mathbf{x}_0, \mathbf{c}))}$, where $R$ is the reward function. The normalized scores serve as rewards in the process of DDPO fine-tuning. DDPO employs proximal policy optimization (PPO) algorithms (Schulman et al., 2017) via importance sampling $\frac{p_\theta(\mathbf{x}_{t-1}|\mathbf{x}_t, \mathbf{c})}{p_{\theta_{old}}(\mathbf{x}_{t-1}|\mathbf{x}_t, \mathbf{c})}$ and clipping. The gradient when applying D-Fusion to DDPO goes as follows.

$$
-\mathbb{E}\left( \sum_{t=1}^{T} \left[ \frac{p_\theta(\mathbf{x}_{t-1}^a \mid \mathbf{x}_t^a, \mathbf{c})}{p_{\theta_{\mathrm{old}}}(\mathbf{x}_{t-1}^a \mid \mathbf{x}_t^a, \mathbf{c})} \nabla_\theta \log p_\theta(\mathbf{x}_{t-1}^a \mid \mathbf{x}_t^a, \mathbf{c}) \hat{r}^a \right. \right.
$$
$$
\left. \left. + \frac{p_\theta(\mathbf{x}_{t-1}^b \mid \mathbf{x}_t^b, \mathbf{c})}{p_{\theta_{\mathrm{old}}}(\mathbf{x}_{t-1}^b \mid \mathbf{x}_t^b, \mathbf{c})} \nabla_\theta \log p_\theta(\mathbf{x}_{t-1}^b \mid \mathbf{x}_t^b, \mathbf{c}) \hat{r}^b \right] \right).
\tag{8}
$$

**Compatibility with DPOK.** Similar to DDPO, DPOK also employs the clipping mechanism in PPO. Meanwhile, DPOK utilizes a value function $V(\mathbf{x}_t, \mathbf{c})$ in their implementation. In our implementation, we replace the value function with reward normalization same as DDPO. Therefore, the gradient when applying D-Fusion to DPOK goes as follows, where $\alpha = 0.99$ and $\beta = 0.01$.

$$
\mathbb{E}\left( \sum_{t=1}^{T} \left[ -\alpha \nabla_\theta \log p_\theta(\mathbf{x}_{t-1}^a \mid \mathbf{x}_t^a, \mathbf{c}) \hat{r}^a \right. \right.
$$
$$
+ \beta \nabla_\theta \mathrm{KL}(p_\theta(\mathbf{x}_{t-1}^a \mid \mathbf{x}_t^a, \mathbf{c}) || p_{\theta_{\mathrm{old}}}(\mathbf{x}_{t-1}^a \mid \mathbf{x}_t^a, \mathbf{c}))
$$
$$
- \alpha \nabla_\theta \log p_\theta(\mathbf{x}_{t-1}^b \mid \mathbf{x}_t^b, \mathbf{c}) \hat{r}^b
$$
$$
\left. \left. + \beta \nabla_\theta \mathrm{KL}(p_\theta(\mathbf{x}_{t-1}^b \mid \mathbf{x}_t^b, \mathbf{c}) || p_{\theta_{\mathrm{old}}}(\mathbf{x}_{t-1}^b \mid \mathbf{x}_t^b, \mathbf{c})) \right] \right).
\tag{9}
$$

---

**Algorithm 1:** Pseudo-code of employing direct preference optimization with D-Fusion for one training round.

---

**Input** :Total denoising timesteps $T$, inner epoch $E$, number of samples each round $N$, prompt list $C$, reward function $R$, pre-trained diffusion model $p_\theta$.

$p_{old} = \text{deepcopy}(p_\theta)$ ;
$p_{old}.\text{require\_grad}(\text{False})$ ;
// Sampling
$D_{sampling} = \{c : [\,] \text{ for } c \text{ in } C\}$ ;
**for** $n \leftarrow 1$ **to** $N$ **do**
     Randomly choose a prompt $c$ from $C$ ;
     Randomly choose $x_T$ from $\mathcal{N}(0, I)$ ;
     Set seed to a random number $s$ ;
     $x_{(T-1):0} = \text{Denoise from } x_T \text{ with } p_\theta \text{ for } T \text{ steps}$ ;
     $r = R(c, x_0)$ ;
     $D_{sampling}[c].\text{append}(\{x_{T:0}, r, s\})$ ;
**end**
// Constructing Visually Consistent Samples
$D_{training} = [\,]$ ;
**for** $c, \{x_{T:0}, r, s\}_{0:K-1} \in D_{sampling}$ **do**
     $D_{temp} = \text{sort } \{x_{T:0}, r, s\}^{1:K}$ in descending order according to $r$ ;
     $D_{reference} = D_{temp}[0 : K//2]$ ;
     $D_{base} = D_{temp}[K//2 : K]$ ;
     **for** $\{x_{T:0}^r, r^r, s^r\}, \{x_{T:0}^b, r^b, s^b\} \in zip(D_{reference}, D_{base})$ **do**
         Set seed to $s^r$ ;
         $A_{T:1}^{cross}, K_{T:1}^r, V_{T:1}^r = \text{Denoise from } x_T^r \text{ with } p_\theta \text{ for } T \text{ steps}$ ;
         Extract mask $M_{T:1}$ from $A_{T:1}^{cross}$ using Eq.(4) ;
         Set seed to $s^b$ ;
         $x_T^a = x_T^b$ ;
         **for** $t \leftarrow T$ **to** $1$ **do**
             $x_{t-1}^a = \text{Denoise from } x_t^a \text{ with } p_\theta$, incorporating $M_t, K_t^r, V_t^r$ using Eq.(5) ;
         **end**
         $D_{training}.\text{append}(\{x_{T:0}^b, x_{T:0}^a, c\})$ ;
     **end**
**end**
// Training
**for** $e \leftarrow 1$ **to** $E$ **do**
     $D = \text{shuffle}(D_{training})$ ;
     with grad ;
     **for** $d \in D$ **do**
         $d = \text{shuffle}(d)$ ;
         **for** $\{x_t^b, x_{t-1}^b, x_t^a, x_{t-1}^a, c\} \in d$ **do**
             update $\theta$ with gradient descent using Eq. (6) ;
         **end**
     **end**
**end**

---

# E. More Experimental Results

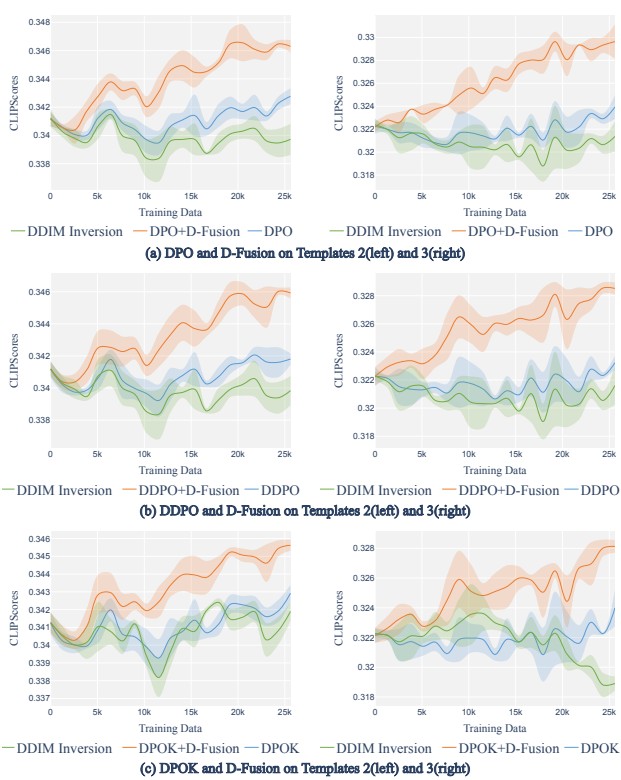

(a) DPO and D-Fusion on Templates 2(left) and 3(right)

(b) DDPO and D-Fusion on Templates 2(left) and 3(right)

(c) DPOK and D-Fusion on Templates 2(left) and 3(right)

*Figure 11.* More ablation studies on denoising trajectories and RL algorithms with templates 2 and 3.

As a supplement to Section 4.4, we also conduct ablation studies on denoising trajectories and RL algorithms on templates 2 and 3, as illustrated in Figure 11. The experimental results show the same conclusion as the ablation studies on template 1. That is, on the one hand, constructing denoising trajectories by DDIM inversion is not an effective approach for RL training. On the other hand, integrating D-Fusion can enhance the effect of different RL algorithms, which can further improve the alignment of diffusion models.

# F. More Samples

In this appendix, we present more samples generated by the diffusion models fine-tuned with visually consistent samples. In detail, Figure 12 shows more samples of our method when compatible with DPO, DDPO and DPOK on template 1. Correspondingly, Figure 13 and Figure 14 show more samples on templates 2 and 3. Moreover, Figure 15 shows more samples when generalized to unseen prompts.

# G. Prompt Lists

We present the prompt lists used in our experiments in this section. Meanwhile, we list the mask thresholds correspond-

ing to each item used in Eq.(4). For each prompt template, we collect 40 prompts for training and another 40 prompts for generalization test. The full prompt lists are shown in Table 4, Table 5 and Table 6.

The mask thresholds are not hyperparameters that require meticulous tuning, and can be predetermined through low-cost methods. In our experiments, we predetermined the thresholds by sampling a few images for each prompt, and assessing whether the chosen threshold value allows the mask to outline corresponding object. This selection process does not require high precision. As shown in the prompt lists, the thresholds we use (predominantly 0.005, 0.01, 0.015, 0.02, and 0.03) are fairly coarse values. To apply our method to new prompts, one can simply sample a few images and determine appropriate thresholds through straightforward observation.

| SD | DPO | DPO+D-Fusion | DDPO | DDPO+D-Fusion | DPOK | DPOK+D-Fusion |

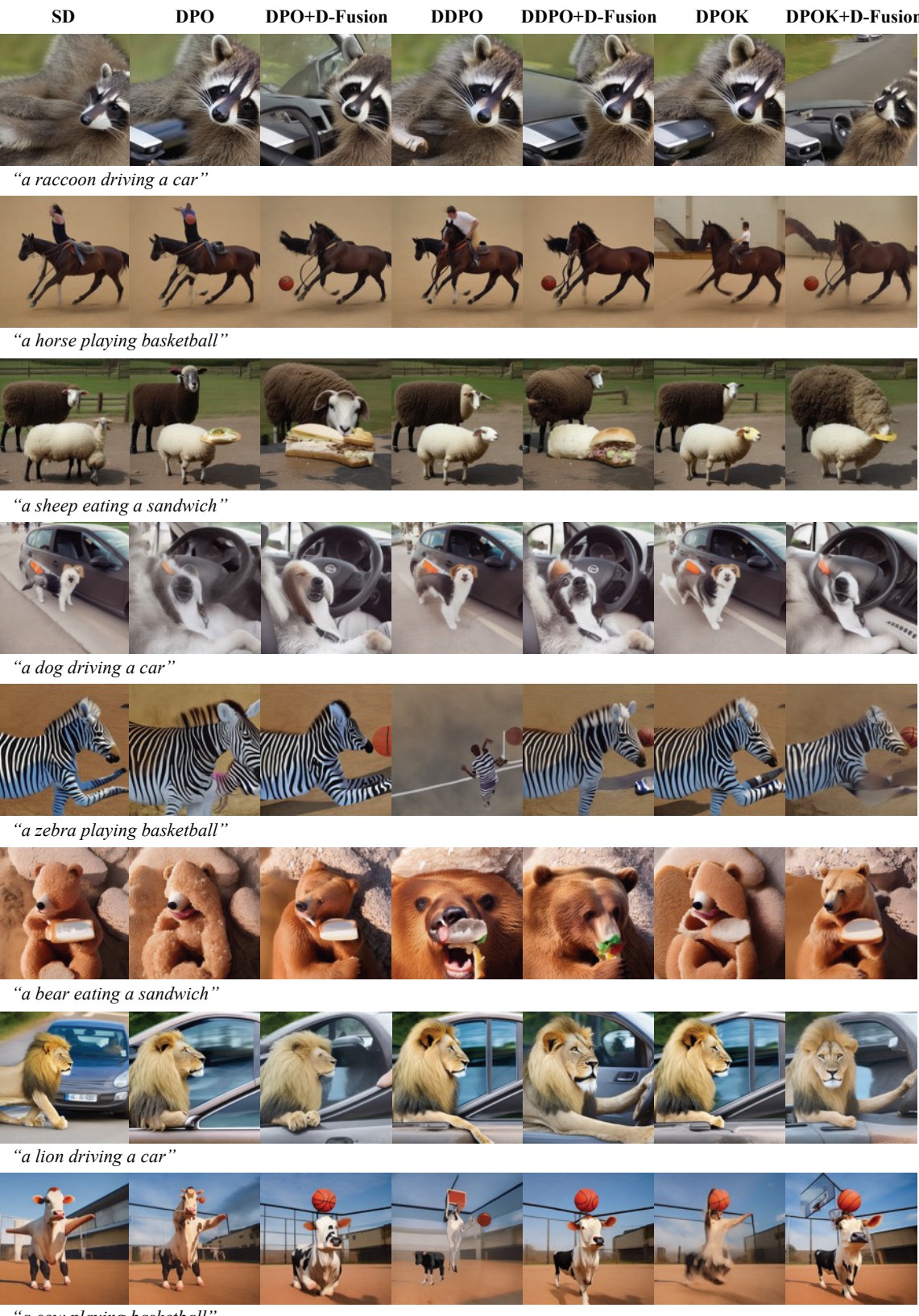

*"a raccoon driving a car"*

*"a horse playing basketball"*

*"a sheep eating a sandwich"*

*"a dog driving a car"*

*"a zebra playing basketball"*

*"a bear eating a sandwich"*

*"a lion driving a car"*

*"a cow playing basketball"*

*Figure 12.* More samples generated by the diffusion models with template 1. The models are fine-tuned by different RL methods with or without D-Fusion.

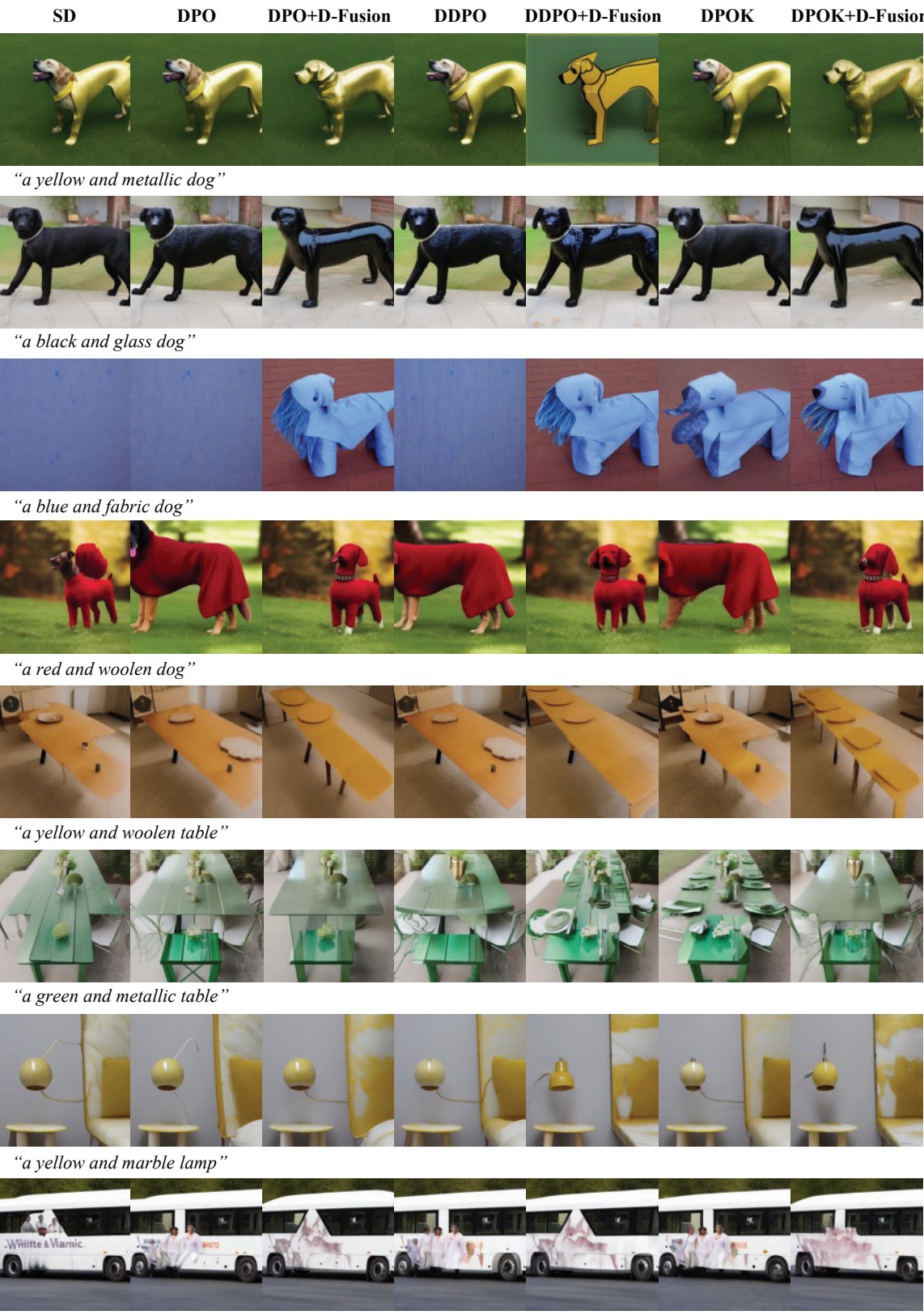

*Figure 13.* More samples generated by the diffusion models with template 2. The models are fine-tuned by different RL methods with or without D-Fusion.

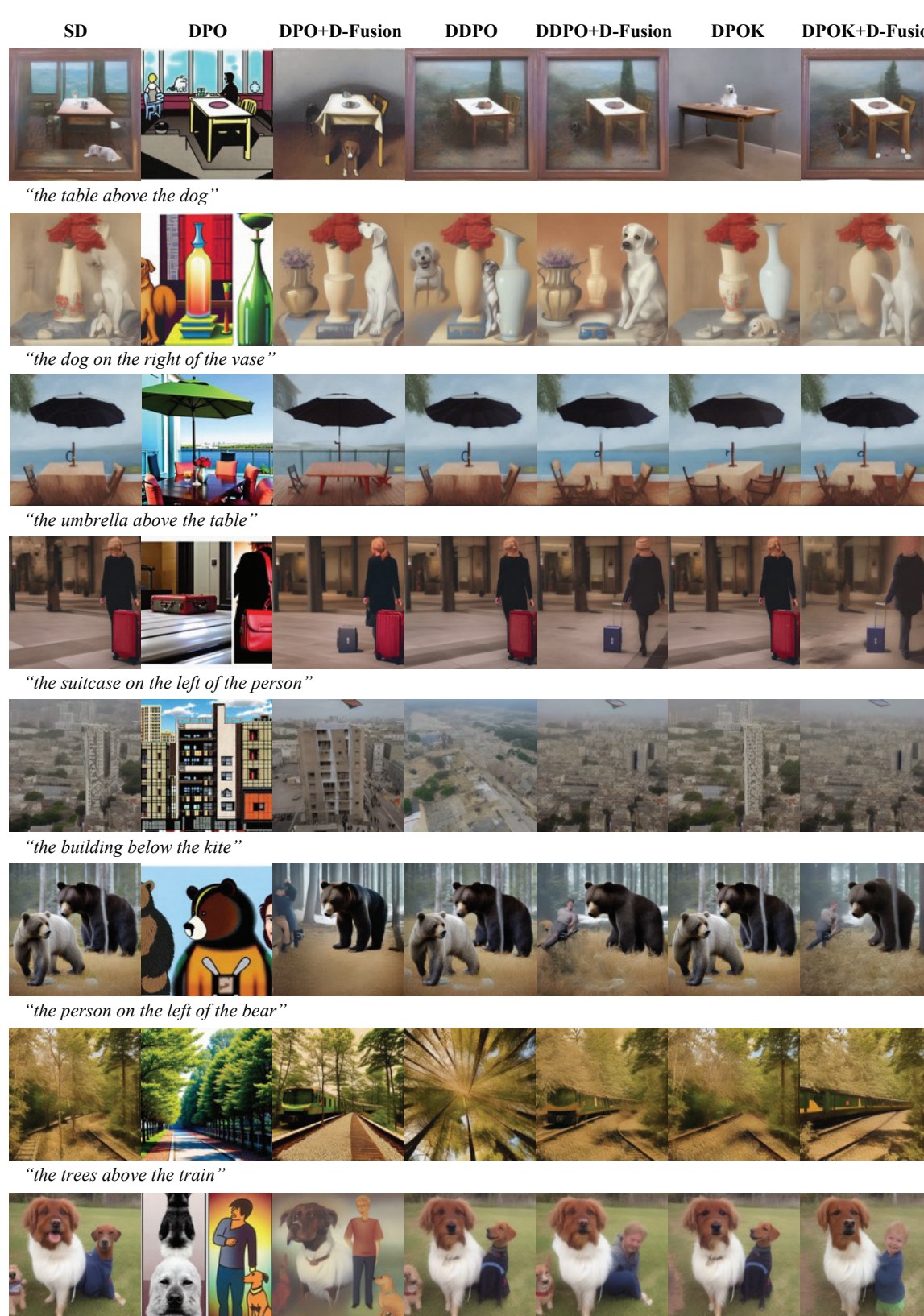

*Figure 14.* More samples generated by the diffusion models with template 3. The models are fine-tuned by different RL methods with or without D-Fusion.

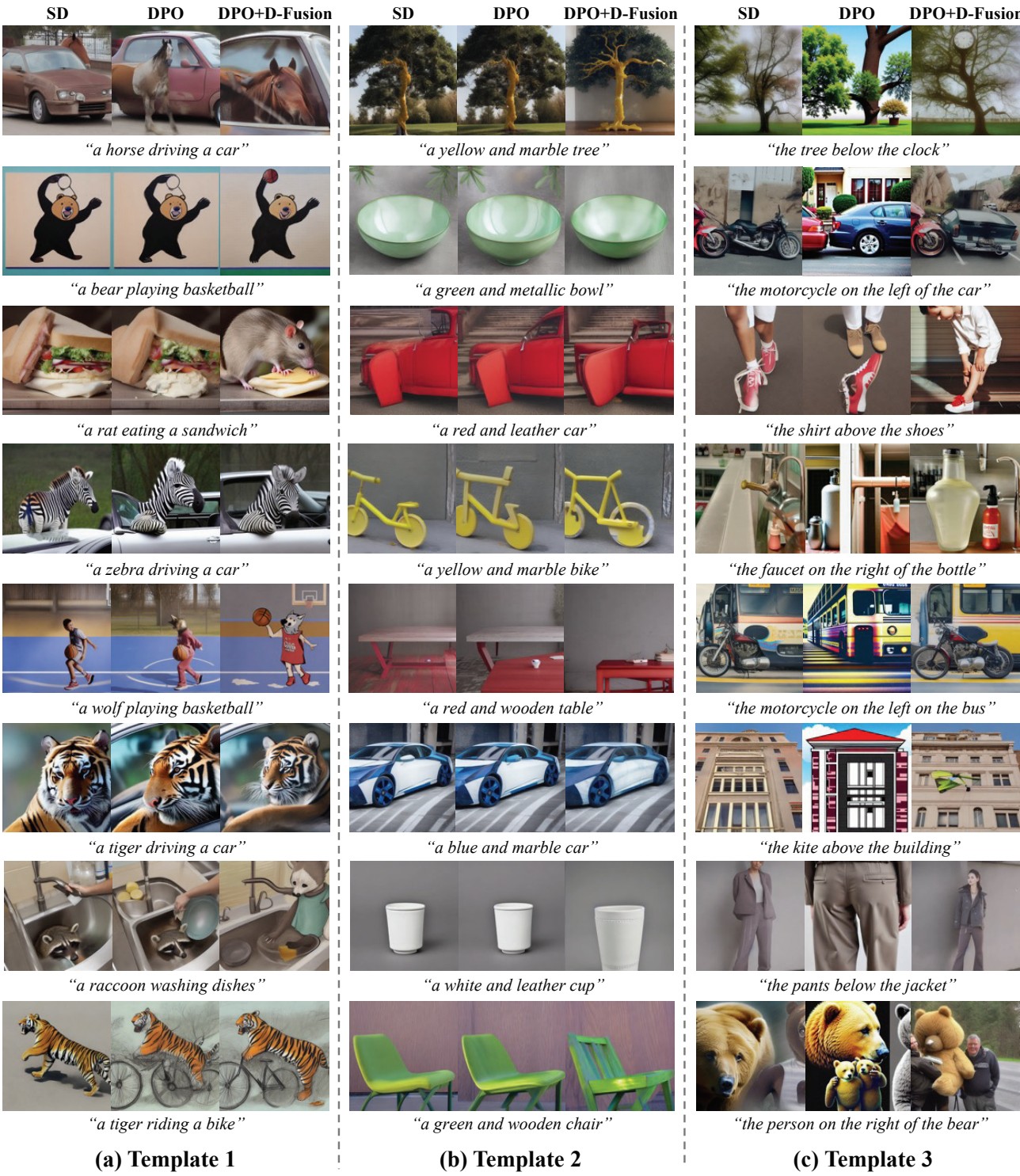

*Figure 15.* More samples when generalized to unseen prompts on templates 1, 2 and 3.

*Table 4.* Prompt lists and mask thresholds for template 1.

| Training list | | Test list |
| --- | --- | --- |
| Prompt | Mask Threshold | Prompt |
| a cat eating a sandwich | 0.03; 0.005 | a cat playing basketball |
| a cat driving a car | 0.03; 0.01 | a dog eating a sandwich |
| a dog driving a car | 0.03; 0.01 | a horse driving a car |
| a dog playing basketball | 0.03; 0.02 | a monkey playing basketball |
| a horse playing basketball | 0.03; 0.02 | a rabbit eating a sandwich |
| a horse eating a sandwich | 0.03; 0.005 | a zebra driving a car |
| a monkey eating a sandwich | 0.03; 0.005 | a sheep playing basketball |
| a monkey driving a car | 0.03; 0.01 | a deer eating a sandwich |
| a rabbit driving a car | 0.03; 0.01 | a cow driving a car |
| a rabbit playing basketball | 0.03; 0.02 | a goat playing basketball |
| a zebra playing basketball | 0.03; 0.02 | a lion eating a sandwich |
| a zebra eating a sandwich | 0.03; 0.005 | a tiger driving a car |
| a sheep eating a sandwich | 0.03; 0.005 | a bear playing basketball |
| a sheep driving a car | 0.03; 0.01 | a raccoon eating a sandwich |
| a deer driving a car | 0.03; 0.01 | a fox driving a car |
| a deer playing basketball | 0.03; 0.02 | a wolf playing basketball |
| a cow playing basketball | 0.03; 0.02 | a lizard eating a sandwich |
| a cow eating a sandwich | 0.03; 0.005 | a shark driving a car |
| a goat eating a sandwich | 0.03; 0.005 | a whale playing basketball |
| a goat driving a car | 0.03; 0.01 | a dolphin eating a sandwich |
| a lion driving a car | 0.03; 0.01 | a squirrel driving a car |
| a lion playing basketball | 0.03; 0.02 | a mouse playing basketball |
| a tiger playing basketball | 0.03; 0.02 | a rat eating a sandwich |
| a tiger eating a sandwich | 0.03; 0.005 | a turtle driving a car |
| a bear eating a sandwich | 0.03; 0.005 | a frog playing basketball |
| a bear driving a car | 0.03; 0.01 | a chicken eating a sandwich |
| a raccoon driving a car | 0.03; 0.01 | a duck driving a car |
| a raccoon playing basketball | 0.03; 0.02 | a goose playing basketball |
| a fox playing basketball | 0.03; 0.02 | a pig eating a sandwich |
| a fox eating a sandwich | 0.03; 0.005 | a llama driving a car |
| a wolf eating a sandwich | 0.03; 0.005 | a lion washing dishes |
| a wolf driving a car | 0.03; 0.01 | a tiger riding a bike |
| a lizard driving a car | 0.03; 0.01 | a bear playing chess |
| a lizard playing basketball | 0.03; 0.02 | a raccoon washing dishes |
| a shark playing basketball | 0.03; 0.02 | a fox riding a bike |
| a shark eating a sandwich | 0.03; 0.005 | a wolf playing chess |
| a whale eating a sandwich | 0.03; 0.005 | a lizard washing dishes |
| a whale driving a car | 0.03; 0.01 | a shark riding a bike |
| a dolphin driving a car | 0.03; 0.01 | a whale playing chess |
| a dolphin playing basketball | 0.03; 0.02 | a dolphin washing dishes |

*Table 5.* Prompt lists and mask thresholds for template 2.

| Training list | | Test list |
|---|---|---|
| Prompt | Mask Threshold | Prompt |
| a black and woolen bus | 0.005 | a red and wooden table |
| a black and plastic dog | 0.03 | a green and metallic bowl |
| a green and metallic table | 0.02 | a white and woolen bowl |
| a black and glass table | 0.02 | a blue and stone sculpture |
| a white and marble bus | 0.03 | a yellow and glass bus |
| a red and woolen dog | 0.01 | a black and woolen sculpture |
| a green and glass lamp | 0.03 | a yellow and fabric table |
| a red and glass dog | 0.015 | a yellow and marble tree |
| a black and glass bowl | 0.03 | a blue and wooden toy |
| a green and stone lamp | 0.03 | a red and plastic sculpture |
| a yellow and wooden tree | 0.02 | a white and leather tree |
| a black and marble vase | 0.03 | a yellow and metallic sculpture |
| a yellow and metallic dog | 0.015 | a white and wooden lamp |
| a yellow and woolen table | 0.005 | a yellow and stone bus |
| a blue and woolen vase | 0.005 | a red and fabric toy |
| a yellow and plastic bus | 0.03 | a green and wooden dog |
| a black and stone sculpture | 0.025 | a white and woolen table |
| a red and marble table | 0.02 | a black and stone table |
| a white and plastic lamp | 0.03 | a white and metallic tree |
| a yellow and leather table | 0.02 | a green and plastic sculpture |
| a red and leather toy | 0.01 | a white and marble car |
| a red and leather table | 0.02 | a white and leather boat |
| a blue and plastic sword | 0.03 | a blue and fabric clock |
| a black and fabric tree | 0.02 | a white and stone chair |
| a yellow and fabric dog | 0.015 | a green and leather chair |
| a black and plastic table | 0.02 | a yellow and wooden cup |
| a white and metallic sculpture | 0.02 | a white and leather cup |
| a black and leather tree | 0.02 | a green and wooden chair |
| a blue and marble toy | 0.015 | a black and wooden car |
| a black and glass dog | 0.015 | a red and wooden plate |
| a black and fabric bus | 0.03 | a red and glass car |
| a green and wooden vase | 0.03 | a yellow and stone car |
| a blue and plastic table | 0.02 | a white and metallic cup |
| a green and fabric dog | 0.015 | a white and stone car |
| a black and stone bowl | 0.02 | a blue and marble car |
| a black and stone tree | 0.02 | a red and woolen bike |
| a black and glass sculpture | 0.015 | a yellow and marble bike |
| a yellow and marble lamp | 0.03 | a blue and wooden chair |
| a blue and fabric dog | 0.015 | a red and marble plate |
| a green and glass sword | 0.03 | a red and leather car |

*Table 6.* Prompt lists and mask thresholds for template 3.

| Training list | | Test list |
|---|---|---|
| Prompt | Mask Threshold | Prompt |
| the umbrella above the table | 0.03; 0.01 | the shirt above the shoes |
| the trees above the train | 0.01; 0.03 | the jacket above the pants |
| the laptop above the table | 0.02; 0.01 | the kite above the building |
| the table below the laptop | 0.01; 0.02 | the kite above the sand |
| the building below the tower | 0.005; 0.01 | the monitor above the keyboard |
| the snowboard below the person | 0.03; 0.005 | the keyboard above the mouse |
| the dog on the right of the vase | 0.01; 0.03 | the glasses below the laptop |
| the table above the dog | 0.01; 0.01 | the shirt above the jeans |
| the shirt above the pants | 0.01; 0.01 | the jeans above the shoes |
| the suitcase above the dog | 0.03; 0.01 | the bag below the sink |
| the suitcase on the left of the person | 0.03; 0.005 | the hat above the sunglasses |
| the dog below the suitcase | 0.01; 0.03 | the tree below the clock |
| the dog on the left of the person | 0.02; 0.005 | the clock above the tree |
| the person on the right of the dog | 0.01; 0.02 | the skis below the pants |
| the person on the right of the suitcase | 0.01; 0.03 | the pants below the jacket |
| the person on the right of the hand | 0.01; 0.01 | the sunglasses below the hat |
| the helmet above the glasses | 0.03; 0.01 | the car on the right of the umbrella |
| the helmet above the person | 0.03; 0.01 | the phone on the right of the monitor |
| the roof above the bus | 0.01; 0.01 | the shirt below the helmet |
| the wheel below the engine | 0.01; 0.005 | the pants below the shirt |
| the engine above the wheel | 0.005; 0.01 | the person on the right of the bear |
| the car on the right of the person | 0.01; 0.01 | the bear on the right of the person |
| the table below the glasses | 0.01; 0.01 | the bear on the left of the person |
| the building below the kite | 0.01; 0.03 | the bus on the right of the car |
| the sand below the kite | 0.03; 0.03 | the bus on the right of the motorcycle |
| the mouse below the keyboard | 0.03; 0.03 | the car on the left of the bus |
| the computer below the counter | 0.01; 0.005 | the motorcycle on the left of the bus |
| the person on the left of the ball | 0.01; 0.01 | the motorcycle on the left of the car |
| the ball on the right of the person | 0.01; 0.01 | the table below the monitor |
| the person on the left of the pillow | 0.01; 0.02 | the person below the monitor |
| the bowl on the right of the plate | 0.01; 0.01 | the basket on the right of the person |
| the building on the right of the truck | 0.01; 0.01 | the faucet on the right of the bottle |
| the person on the left of the bottle | 0.01; 0.02 | the pot on the right of the faucet |
| the bottle on the right of the person | 0.02; 0.01 | the van on the right of the car |
| the box on the left of the post | 0.01; 0.01 | the car on the left of the van |
| the truck on the right of the car | 0.01; 0.01 | the person on the left of the train |
| the jacket on the left of the coat | 0.01; 0.01 | the hat on the left of the shirt |
| the monitor on the left of the person | 0.01; 0.01 | the plate on the left of the glasses |
| the phone on the left of the person | 0.01; 0.01 | the person on the left of the cart |
| the person on the left of the bear | 0.01; 0.03 | the bed on the left of the chair |

