# OpenReview forum: "D-Fusion: Direct Preference Optimization for Aligning Diffusion Models with Visually Consistent Samples"
_ICML.cc/2025/Conference — ICML 2025 poster_

### Official Review · Reviewer_ri7A · 2025-03-10

**Overall Recommendation:** 4

**Summary:**

This paper addresses the issue the effectiveness of preference learning (e.g. Direct Preference Optimization (DPO)) for text-to-image diffusion model is limited by the visual inconsistency between the sample that aligns better with the text and the sample that aligns less with the prompt, making it difficult to focus on the relevant differences between the positive and negative samples. The authors introduce D-Fusion (Denoising Trajectory Fusion), a method to construct visually consistent samples for preference learning. D-Fusion works by performing mask-guided self-attention fusion between a well-aligned reference image and a poorly-aligned base image, resulting in a target image that is both well-aligned and visually consistent with the base image, while also retaining the denoising trajectories necessary for DPO training. Through comprehensive experiments on Stable Diffusion, the authors demonstrate that applying D-Fusion can significantly improve prompt-image alignment when used with different reinforcement learning algorithms like DPO, DDPO, and DPOK. The paper highlights the necessity of using visually consistent image pairs for fine-tuning diffusion models with DPO and introduces D-Fusion as a compatible approach to address this challenge

**Claims And Evidence:**

The claims made in the submission appear to be supported by evidence. Both qualitative results (Figure 4) and quantitative results (Figure 5, Table 1), including CLIPScore metrics and human preference tests (Figure 6), consistently demonstrate the effectiveness of D-Fusion in improving text-image alignment, compared to the plain DPO and the base model. These support the hypothesis that preference optimization can work better with visual similar pairs, and demonstrate the effectiveness of the proposed method.

**Essential References Not Discussed:**

I'm not aware of significant missing references.

**Experimental Designs Or Analyses:**

The experiment designs are pretty standard for evaluating text-to-image diffusion models. On the other hand, I'm not very sure I understand the motivation and the value for having the comparison with DDIM inversion in ablation study (or whether it should be called an ablation studies). I'd like to see more details about that.

**Methods And Evaluation Criteria:**

Denoising Trajectory Fusion is an empirical method, motivated with intuition.

It aims to generate well-aligned and also visually similar positive image samples for preference learning. The process involves two key steps:
1. Cross-Attention Mask Extraction: A mask is extracted from the cross-attention maps of the reference image's denoising process. This mask identifies the image regions corresponding to the prompt's content.
2. Self-Attention Fusion: Starting with the same random noise as the base image, a target image is denoised. During this process, the self-attention keys and values of the target image are fused with those of the reference image using the extracted mask. In the masked (prompt-related) areas, the target image adopts the self-attention information from the reference image, enhancing alignment. In the unmasked areas, it retains its own information, ensuring visual consistency with the base image.

Although there is no theoretical support, the intuition makes sense and the evaluation criteria used in the paper are standards for evaluating text-to-image diffusion models.

**Other Comments Or Suggestions:**

- Writing quality has room to improve.
- (Line 25, abstract) On the one hand -> On one hand
- I suggest using $I^t$ instead of $I^a$ for the target image, following $I^b$ for base image and $I^r$ for reference image.

**Other Strengths And Weaknesses:**

No additional comments.

**Questions For Authors:**

As mentioned, can you expand the ablation study section to provide more clarity on the motivation and the value for having the comparison with DDIM inversion?

**Relation To Broader Scientific Literature:**

It's related the literature of text-to-image diffusion model alignment.

**Theoretical Claims:**

This is a pure empirical paper.

---

> ### Author Rebuttal · Authors · 2025-04-01
>
> Thanks a lot for your time and efforts in reviewing our paper. We are happy that you think the intuition makes sense. In addition, we feel pleased that you approve our experiment design.
>
> Below are responses to your concerns and suggestions. Please let us know if you require any further information, or if anything is unclear.
>
> ## 1. Clarification on Comparison with DDIM Inversion
>
> > As mentioned, can you expand the ablation study section to provide more clarity on the motivation and the value for having the comparison with DDIM inversion?
>
> Thanks for this concern! We would like to clarify on it in a clearer manner.
>
> Firstly, fine-tuning with reinforcement learning requires access to the **denoising trajectories** of the images. This paper highlights the necessity of fine-tuning diffusion models using visually consistent samples. However, constructing visually consistent samples through **image editing** is not feasible, as the editing process disrupts the denoising trajectory (See right column of Line 90 in the paper for detail). The advantage of D-Fusion lies in its ability to generate visually consistent samples while **preserving their denoising trajectories**.
>
> However, there exist two straightforward methods to create a denoising trajectory for any given image: the **forward process** of diffusion models and the **DDIM inversion**. (The former simply adds noise to an image, which is so trivial that we did not mention it in the paper.) The created denoising trajectories can also be used for reinforcement learning fine-tuning.
>
> Therefore, we designed this comparative experiment, where the training data consists of the **same visually consistent samples**, but the **denoising trajectories come from D-Fusion and DDIM inversion respectively**. The results indicate that training with denoising trajectories created by DDIM inversion performs poorly. This effectively rules out the method of fine-tuning the model using DDIM Inversion through the following steps:
>
> * generating visually consistent samples via **image editing**,
> * creating denoising trajectories using **DDIM inversion**,
> * fine-tuning with samples and trajectories.
>
> Thus, this comparison further demonstrates the advantage of D-Fusion (*i.e.*, generating visually consistent samples while preserving their denoising trajectories).
>
> We would like to further discuss the **forward process**. Adding noise to images through the **forward process** essentially turns the training to supervised fine-tuning (SFT). We have also provided experimental results on SFT. if you're interested, please refer to Figure 1(a) at this link (https://anonymous.4open.science/r/ICML-25-Rebuttal-2703/rebuttal_2703.pdf). As shown, SFT also performs poorly in improving alignment, further demonstrating the effectiveness of RL training with D-Fusion.
>
> ## 2. About Writing Suggestions
>
> > * Writing quality has room to improve.
> > * (Line 25, abstract) On the one hand -> On one hand.
> > * I suggest using $I^t$ instead of $I^a$ for the target image, following $I^b$ for base image and $I^r$ for reference image.
>
> Thank you again for providing these suggestions. In the final version, we will conduct a **thorough review** to correct errors and enhance writing quality. For instance, we will revise the phrase "on the one hand" in the abstract, as you pointed out, to "on one hand". We will also revise $I^a$ to $I^t$, in order to keep consistent with $I^r$ and $I^b$.

---

### Official Review · Reviewer_n8VD · 2025-03-10

**Overall Recommendation:** 3

**Summary:**

This paper introduces **D-Fusion**, a method for improving text-image alignment in diffusion models via **Direct Preference Optimization (DPO)**. It identifies a key challenge in prior DPO approaches—**visual inconsistency** between well-aligned and poorly-aligned images—which hinders the model's ability to learn effective alignment strategies. To address this, D-Fusion **constructs visually consistent training samples** using **self-attention fusion** while preserving denoising trajectories, ensuring compatibility with RL-based training. The paper demonstrates **significant improvements in prompt-image alignment** over naive DPO on Stable Diffusion (SD) 2.1 across various evaluation metrics, including CLIPScore and human preference tests. The method is also shown to be **compatible with multiple RL algorithms**, including DPO, DDPO, and DPOK.

**Claims And Evidence:**

The paper's central claims are supported by extensive empirical results:

1. **D-Fusion improves text-image alignment**: Experiments demonstrate superior CLIPScore performance and higher human preference rates compared to standard DPO.
2. **D-Fusion constructs visually consistent training samples**: Qualitative examples and ablation studies confirm that self-attention fusion effectively preserves alignment features while ensuring visual consistency.
3. **D-Fusion generalizes across different RL methods**: The results indicate that integrating D-Fusion with different RL fine-tuning approaches (DPO, DDPO, DPOK) leads to improved alignment.

However, one limitation is that all experiments are conducted on **SD 2.1**, an older model, without evaluation on **more advanced diffusion architectures** such as **SDXL** or **Flux**, which may better reflect the state of the field.

**Essential References Not Discussed:**

**SDXL and Modern Diffusion Architectures**

**Experimental Designs Or Analyses:**

The experimental setup is **mostly sound**, but there are some limitations:

1. **Model Selection Bias**: The choice of SD 2.1 for evaluation is outdated, raising concerns about applicability to modern architectures.
2. **Lack of Baselines Beyond DPO Variants**: While the paper evaluates multiple DPO-based methods, it does not compare against alternative alignment techniques (e.g., **reward models, preference learning in SDXL**).
3. **Generalization to Real-World Tasks**: The evaluation primarily focuses on synthetic prompt-image pairs, with no discussion of real-world generative applications (e.g., inpainting, video synthesis).

**Methods And Evaluation Criteria:**

The method is well-motivated and addresses a real challenge in diffusion model alignment. The **benchmarking approach is reasonable**, using:

- **CLIPScore** as a quantitative metric
- **Human preference testing** as a qualitative evaluation
- **Three types of prompts (actions, attributes, spatial relationships)** to ensure diversity

One concern is **the absence of comparison to state-of-the-art models** such as SDXL or Flux. Given the rapid advancements in diffusion models, it is unclear whether the improvements hold in **larger-scale architectures**.

**Other Comments Or Suggestions:**

1. **Why was SD 2.1 chosen instead of SDXL?**
2. **Does D-Fusion improve alignment in other generative tasks (e.g., inpainting, video synthesis)?**
3. **How does D-Fusion compare to reward-based RL approaches for alignment?**
4. **What is the computational overhead of self-attention fusion?**

**Other Strengths And Weaknesses:**

### **Strengths:**

1.  **Well-motivated problem** (prompt-image misalignment)
2.  **D-Fusion is simple yet effective** (self-attention fusion for visual consistency)
3.  **Extensive evaluation** (multiple RL algorithms, CLIPScore, human preference)

### **Weaknesses:**

1.  **Choice of model (SD 2.1) is outdated** → Lacks validation on SDXL/Flux
2.  **No comparison to alternative fine-tuning approaches** (e.g., LoRA for alignment, reward-based RLHF)
3.  **Does not explore computational efficiency** (D-Fusion adds complexity—does it scale?)

**Questions For Authors:**

1. **How does mask-guided fusion affect diversity?**
   - Does it introduce mode collapse (favoring specific visual patterns)?
2. **Are there failure cases?**
   - Are there scenarios where D-Fusion reduces prompt-image alignment instead of improving it?

**Relation To Broader Scientific Literature:**

The work aligns with ongoing research in **reinforcement learning for generative models** and **diffusion model alignment**. Specifically, it builds on:

- **DPO-based fine-tuning for LLMs and diffusion models** (Wallace et al., 2023; Fan et al., 2023)
- **Attention control techniques for image editing** (Cao et al., 2023; Hertz et al., 2022)
- **CLIP-based alignment evaluation** (Radford et al., 2021)

However, the paper does not **explicitly compare to recent SDXL-related studies** or Flux-based advancements, which limits its impact.

**Theoretical Claims:**

The paper primarily focuses on empirical improvements rather than theoretical derivations. The **formulation of diffusion models as a sequential decision-making process** and the **DPO objective function** appear correct and align with prior work.

---

> ### Author Rebuttal · Authors · 2025-04-01
>
> Thanks for providing these detailed and helpful comments! We appreciate that you think our method well-motivated and approve of our experiments.
>
> Responses to your concerns are as follows. We provide additional experimental results **at this link** (https://anonymous.4open.science/r/ICML-25-Rebuttal-2703/rebuttal_2703.pdf).
>
> ## 1. About Base Model
>
> Thanks for recommending advanced models like SDXL. In fact, we have tried to use them. Unfortunately, due to limited computational resources (24GB 3090/4090 GPU), we are unable to fine-tune them and can only perform inference.
>
> In Figure 6 and 7 of the above link, we show some images generated using D-Fusion on **SDXL** and **SD3.5**. As shown, D-Fusion is capable of producing well-aligned and visually consistent samples with these models. We suggest that fine-tuning with these samples still has the potential to enhance alignment of these models.
>
> We chose SD2.1 as it is the most advanced model our resources allow. Notably, much influential related work conducted experiments on SD1.4 and SD1.5 [1,2,3]. We will appreciate your understanding regarding our limitation of base model.
>
> ## 2. More Baselines
>
> > No comparison to alternative approaches (e.g., LoRA for alignment, reward-based RLHF)
>
> Thanks for your suggestion. We would like to clarify that DDPO and DPOK are forms of reward-based RLHF. As follows, we provide a categorized list of the methods compared in our experiments, with **bold** ones indicating those newly added during rebuttal.
>
> * SFT: **Lora Fine-tuning**
> * Reward-based RLHF: DDPO, DPOK, **Policy Gradient**
> * DPO: DPO, **DenseReward**[4]
>
> Results of added experiments are shown in Figure 1 of the link. D-Fusion are well compatible with these methods. As shown, incorporating D-Fusion leads to greater alignment improvement.
>
> If there exist methods you believe should be included, please don't hesitate to tell us!
>
> ## 3. Real-World Task
>
> > Real-world applications (e.g., inpainting, video synthesis).
>
> Thanks for your suggestion. To the best of our knowledge, inpainting and video synthesis are rarely explored in previous RL-based studies. Given the time constraints of rebuttal period, we were unable to implement them well. However, RL-based fine-tuning follows an **end-to-end** training paradigm, where simply modifying reward function allows adaptation to various downstream tasks. We believe that RL-based methods hold great potential to benefit these real-world tasks.
>
> Additionally, we conducted experiments on tasks that are commonly studied in prior work, including improving **human preference** and **aesthetic quality**. As shown in Figure 2 of the link, D-Fusion can enhance the performance of RL fine-tuning on these tasks. Hope these tasks are sufficient to demonstrate the generalization ability of our method.
>
> ## 4. Computational Overhead
>
> Thanks for this concern. We provide a detailed analysis in response to reviewer tsGd under “*3. Clarification on Efficiency*”. We would appreciate it if you could take a look.
>
> In our experiments, DPO takes 78 min per round, whereas D-Fusion extends it to 91 min. The additional overhead is affordable.
>
> ## 5. About Diversity and Failure Cases
>
> > How does fusion affect diversity? Are there failure cases?
>
> Thanks for these questions!
>
> In fact, D-Fusion can **mitigate loss of diversity** caused by RL. It is widely recognized that RL may cause the models to only generate images of specific patterns [1]. For instance, when using template "*a(n) [animal] [human activity]*", cartoon images tend to receive high alignment scores, because these prompts are often depicted in cartoon styles in the pre-training data. During RL, the model may learn to generate only cartoon images to achieve high alignment, which reduces diversity.
>
> D-Fusion helps mitigate this issue. The visually consistent sample pairs constructed by D-Fusion typically adopt **similar styles**, but receive **different alignment scores**. When training with these samples, the model does not associate a specific style with high alignment, thus maintaining diversity.
>
> In Table 1 of the link, we show the diversity of images generated by fine-tuned models. As shown, employment of D-Fusion helps alleviate reduction in diversity.
>
> D-Fusion indeed occasionally results in **failure cases**. We show some cases in Figure 5 of the link. As observed, the failures likely arise from **insufficient attention** given to certain objects when sampling, causing the cross-attention mask to poorly outline their positions. This leads to confusion during subsequent fusion. However, based on our experimental results, the presence of a small number of failure cases does not significantly impact fine-tuning performance.
>
> [1] Training Diffusion Models with Reinforcement Learning.
>
> [2] DPOK: Reinforcement Learning for Fine-tuning Text-to-Image Diffusion Models.
>
> [3] Aligning Text-to-Image Models using Human Feedback.
>
> [4] A Dense Reward View on Aligning Text-to-Image Diffusion with Preference.

---

### Official Review · Reviewer_tsGd · 2025-03-13

**Overall Recommendation:** 3

**Summary:**

This paper introduces D-Fusion, a novel approach to addressing the misalignment between generated images and their corresponding text prompts. D-Fusion constructs DPO-trainable visually consistent samples to mitigate the visual inconsistency present in previous DPO methods. First, the mask-guided self-attention fusion ensures that the generated images are not only well-aligned with the text prompts but also visually consistent with given poorly aligned images. Second, D-Fusion preserves the denoising trajectories of the resulting images, facilitating effective DPO training. Experimental results demonstrate significant improvements in prompt-image alignment.

**Claims And Evidence:**

Most are supported, but I believe there is room for improvement.

**Essential References Not Discussed:**

No.

**Experimental Designs Or Analyses:**

No issue.

**Methods And Evaluation Criteria:**

Yes.

**Other Comments Or Suggestions:**

No.

**Other Strengths And Weaknesses:**

### **Strengths**

- The paper is well-written and easy to follow.
- The motivation is clear, and the proposed method is solid.
- The experiments are comprehensive and effectively demonstrate the effectiveness of D-Fusion.

### **Weaknesses**

- In D-Fusion, cross-attention mask extraction relies on findings from SD’s cross-attention maps, which limits its applicability to other diffusion architectures, such as DiT.
- The prompts used in this paper are simple, making it unclear whether D-Fusion can perform well with more complex prompts.
- The efficiency of D-Fusion has not been evaluated.

**Questions For Authors:**

See “Weaknesses”.

**Relation To Broader Scientific Literature:**

D-Fusion is related to the diffusion models and DPO. The cross-attention mask extraction of D-Fusion is inspired by [1] and [2].

[1] Hertz, et al. Prompt-to-prompt image editing with cross attention control.

[2] Cao, et al. Masactrl: Tuning-free mutual self-attention control for consistent image synthesis and editing.

**Theoretical Claims:**

No theoretical claims.

---

> ### Author Rebuttal · Authors · 2025-04-01
>
> We sincerely appreciate your time and effort in reviewing our paper and providing well-structured comments. We are delighted that you think our paper well-written and approve of our motivation and experiments.
>
> Below are our responses to your concerns. Additional experimental results can be found **at this link** (https://anonymous.4open.science/r/ICML-25-Rebuttal-2703/rebuttal_2703.pdf). If anything remains unclear or if you need further information, please feel free to let us know!
>
> ## 1. Applicability to Other Architecture
>
> > (Weakness 1) In D-Fusion, cross-attention mask extraction relies on findings from SD’s cross-attention maps, which limits its applicability to other diffusion architectures, such as DiT.
>
> Many thanks for this concern.
>
> As you pointed out, many diffusion models no longer adopt the U-Net architecture. We take SD3.5, a representative example that uses DiT, as a case study. In SD3.5, there is no explicit cross-attention layer. However, in the self-attention layers, the prompt embeddings are concatenated with image features and processed together, effectively forming an **implicit cross-attention mechanism**. This allows us to extract cross-attention masks and further apply D-Fusion. Examples of this can be found in Figure 7 at the provided link.
>
> More broadly, any model that **incorporates prompt guidance through attention mechanism** could potentially benefit from D-Fusion. However, since many of these models have only gained research attention in recent months, a more in-depth exploration will be pursued in our future work.
>
> ## 2. Complex Prompts
>
> > (Weakness 2) The prompts used in this paper are simple, making it unclear whether D-Fusion can perform well with more complex prompts.
>
> Thank you for this question!
>
> We have observed that previous studies primarily fine-tuned models on short prompts [1,2], with some even fine-tuning on simple labels [3]. To ensure a fair comparison, we followed them to use simple prompts in our experiment.
>
> Moreover, we carefully designed **three types of prompt templates**, which considers the behavior of the object, the attribute of the object and positional relationship between objects respectively. We suggest that these prompts are representative, as most complex prompts can be decomposed into these three fundamental types.
>
> Despite this, we appreciate your reasonable suggestion and have conducted **additional experiments using complex prompts**. As shown in Figure 4(left) at the provided link, D-Fusion can also construct well-aligned visually consistent samples for complex prompts. Furthermore, as demonstrated in Figure 4(right), fine-tuning with visually consistent samples also leads to improved performance under complex prompt conditions.
>
> [1] Training Diffusion Models with Reinforcement Learning.
>
> [2] DPOK: Reinforcement Learning for Fine-tuning Text-to-Image Diffusion Models.
>
> [3] Aligning Text-to-Image Models using Human Feedback.
>
> ## 3. Clarification on Efficiency
>
> > (Weakness 3) The efficiency of D-Fusion has not been evaluated.
>
> Thank you for this suggestion.
>
> We are willing to first analyze the computational overhead of D-Fusion and naive DPO theoretically. As shown in Figure 2 of the paper, each fine-tuning round of naive DPO consists of three steps: **sampling**, **evaluation**, and **training**. Among them, **evaluation** is conducted using AI models, which are highly efficient and can run concurrently with **sampling**. Thus, the total time required for one round in naive DPO can be expressed as $nT_1+nT_2$, where $n$ is the number of images per round, $T_1$ is the time needed to sample each image, and $T_2$ is the time required for training on each image.
>
> D-Fusion introduces an additional **fusion** step following the **evaluation** step. During **fusion** step, half of the images (*i.e.*, those that are well-aligned) are resampled to extract their $K,V$ and mask values, while the remaining half (*i.e.*, those that are poorly aligned) undergo sampling with attention fusion. Consequently, the time required per round in DPO+D-Fusion can be expressed as $nT_1+\frac{n}{2}(T_1+T_1')+nT_2$, where $T_1'$ represents the time needed to apply attention fusion. (Alternatively, storing $K,V$ and mask values during the **sampling** step could eliminate the need for repeated resampling, reducing the total time to $nT_1+\frac{n}{2}T_1'+nT_2$. But we do not recommend this approach, as storing $K,V$ and mask values for all the $n$ images would require excessive memory.)
>
> The computational overhead introduced by D-Fusion is **affordable**. In our experiments, the ratio of processing times is observed as $T_1:T_1':T_2=6:7:33$. This indicates that the majority of the time is spent on training, thus the computational cost of D-Fusion is acceptable. Specifically, when fine-tuning on an RTX 3090 GPU, naive DPO takes 78 minutes per round, whereas D-Fusion extends this to 91 minutes per round.

---

> > ### Comment · Reviewer_tsGd · 2025-04-03
> >
> > Thank you for your response.
> > I believe that has addressed most of my concerns.
> > After considering the reviews from other reviewers, I will maintain my positive score (3: Weak accept).

---

> > > ### Author Response · Authors · 2025-04-04
> > >
> > > We sincerely thank you for these valuable comments and constructive suggestions. And we truly appreciate the time and effort you dedicated to reviewing our work!

---

### Official Review · Reviewer_UTBY · 2025-03-14

**Overall Recommendation:** 2

**Summary:**

This paper focuses on improving DPO for fine-tuning diffusion models. The key problem identified is visual inconsistency in training samples, which hampers the effectiveness of reinforcement learning-based fine-tuning methods like DPO.

**Claims And Evidence:**

The claim that "D-Fusion works across different RL algorithms" is supported primarily through experiments on DPO, DDPO, and DPOK. While these are relevant methods, additional RL-based techniques could be explored to further validate generalizability.

**Essential References Not Discussed:**

The paper does not compare D-Fusion to other Diffusion-PO methods. To name a few:

[1] Diffusion-NPO: Negative Preference Optimization for Better Preference Aligned Generation of Diffusion Models. ICLR 2025.

[3] Diffusion-RPO: Aligning diffusion models through relative preference optimization. arXiv:2406.06382

[4] SePPO: Semi-Policy Preference Optimization for Diffusion Alignment. arXiv:2410.05255

**Experimental Designs Or Analyses:**

**Comparison.**

1. Lack of comparison with DPO-like methods. See *Essential References Not Discussed*.

2. Lack of comparison with SFT.

3. Lack of comparison with RL methods, such PPO and PG. But this is not necessary.

**Ablation Study.** There is no ablation study for the mask threshold hyperparameter.

**Methods And Evaluation Criteria:**

Several meaningful benchmarks are not evaluated, such as Pick-a-Pic, HPSv2, Image Reward, and Aesthetic score.

**Other Comments Or Suggestions:**

N/A

**Other Strengths And Weaknesses:**

The notations are clear and well written.

**Questions For Authors:**

How sensitive is D-Fusion to the mask threshold hyperparameter?

**Relation To Broader Scientific Literature:**

Lack of discussion on other Diffusion DPO variants.

**Theoretical Claims:**

There is no formal proof for D-Fusion, e.g., the optimality of D-Fusion.

---

> ### Author Rebuttal · Authors · 2025-04-01
>
> Thank you for reviewing our paper and providing a lot of suggestions! We appreciate your recognition of our paper's clarity and your time reviewing the paper and supplementary materials in detail.
>
> Responses to your suggestions are as follows. We provide additional experimental results **at this link** (https://anonymous.4open.science/r/ICML-25-Rebuttal-2703/rebuttal_2703.pdf). Please let us know if you require any further information, or if anything is unclear.
>
> ## 1. More Benchmarks
>
> > Several meaningful benchmarks are not evaluated, such as Pick-a-Pic, HPSv2, Image Reward, and Aesthetic score.
>
> Thank you for recommending these meaningful evaluation metrics.
>
> We would like to first clarify that our work focuses on addressing the issue of **prompt-image misalignment**, which is a crucial challenge in controllable generation for diffusion models. The metrics you recommended are designed to assess **human preference** or **aesthetic quality**, which are not the primary tasks our work directly targets.
>
> Nevertheless, we have conducted additional experiments using these metrics you recommended. The results are presented in Figure 2 at the provided link. As shown, when using these metrics as reward models, D-Fusion can still enhance the performance of the RL fine-tuning. This further demonstrates the effectiveness of D-Fusion.
>
> ## 2. Ablation Study on Mask Threshold
>
> > There is no ablation study for the mask threshold hyperparameter.
> > How sensitive is D-Fusion to the mask threshold hyperparameter?
>
> Thank you for raising this important question.
>
> The mask threshold is **not a hyperparameter that requires meticulous tuning**, and can be predetermined through low-cost methods. In our experiments, we predetermined the threshold by sampling a few images for each prompt, and assessing whether the chosen threshold value allows the mask to outline corresponding object. This selection process does not require high precision. As shown in Appendix G, the thresholds we use (predominantly 0.005, 0.01, 0.015, 0.02, and 0.03) are fairly coarse values. If users wish to apply our method to new prompts, they simply need to sample a few images and determine an appropriate threshold through staightforward observation.
>
> Empirically, our method can consistently deliver **stable improvements** even with less precise thresholds. We performed an ablation study on the mask threshold by setting it to 0.002, 0.005, 0.01, 0.02, 0.03 and 0.05, respectively. Each threshold is uniformly applied to **all** prompts in the list. And then we fine-tune the model. The results are shown in Figure 3 at the provided link. As the threshold increases, the fine-tuning performance initially improves and then declines. For thresholds of 0.002 and 0.05, the masks are too inaccurate, leading to a significant drop in performance. In contrast, for threshold values from 0.005 to 0.03, our method can **stably** improve the fine-tuning effect. The experimental results demonstrate the robustness of our method to threshold setting.
>
> ## 3. More Baselines
>
> > * Lack of comparison with DPO-like methods.
> >   * Diffusion-NPO.
> >   * Diffusion-RPO.
> >   * SePPO.
> > * Lack of comparison with SFT.
> > * Lack of comparison with RL methods, such as PPO and PG.
>
> Thank you for pointing out these valuable methods for comparison.
>
> * For the comparison with **SFT**, we have added this experiment.
> * For the comparison with **RL methods**, DDPO is a PPO-based diffusion model fine-tuning method [1], and we have already presented its results in the paper. Additionally, we conducted a comparison with PG.
> * For the comparison with **DPO-like methods**, we have carefully read the papers you recommended. However, we regret to find that none of them have released their implementation code (despite providing repository links), making it difficult for us to verify their details. Nevertheless, we would like to discuss them in the related work section of the final version. Additionally, We conducted a review of existing works, and added a new comparison with DenseReward [2].
>
> The experimental results are shown in Figure 1 at the provided link. D-Fusion can be well **compatible** with SFT, PG and DenseReward. As shown, incorporating D-Fusion into these methods consistently leads to better fine-tuning performance than the original methods.
>
> [1] Training Diffusion Models with Reinforcement Learning.
>
> [2] A Dense Reward View on Aligning Text-to-Image Diffusion with Preference.

---

### Decision · Program_Chairs · 2025-05-01

**Decision:**

Accept (poster)

**Comment:**

The paper introduces D-Fusion, a method for improving text-image alignment in diffusion models by generating visually consistent training pairs. Reviewers found the approach well-motivated, empirically strong, and broadly compatible with existing RL fine-tuning methods.

While one reviewer noted missing baselines and limited theoretical grounding, the majority considered the method practical and clearly presented. Given that the rebuttal effectively addressed most concerns, the reviewers and ACs recommend acceptance. Congratulations to the authors on the good work!